# The Earth system model CLIMBER-X v1.0 – Part 2: The global carbon cycle

Matteo Willeit[1], Tatiana Ilyina[2], Bo Liu[2], Christoph Heinze[3], Mahé Perrette[1], Malte Heinemann[4], Daniela Dalmonech[5], Victor Brovkin[2,6,1], Guy Munhoven[7], Janine Boerker[8], Jens Hartmann[8], Gibran Romero Mujalli[8], and Andrey Ganopolski[1]

[1]Potsdam Institute for Climate Impact Research (PIK), Member of the Leibniz Association, P.O. Box 6012 03, D-14412 Potsdam Germany
[2]Max Planck Institute for Meteorology, Hamburg, Germany
[6]CEN, University of Hamburg, Germany; also a guest at PIK, Potsdam, Germany
[3]University of Bergen, Bergen, Norway
[4]Christian-Albrechts-Universität zu Kiel, Kiel, Germany
[5]Institute for Agriculture and Forestry Systems in the Mediterranean, National Research Council of Italy (CNR-ISAFOM), Perugia, Italy
[7]Dépt. d'Astrophysique, Géophysique et Océanographie, Université de Liège, Liège, Belgium
[8]Universität Hamburg, Hamburg, Germany

**Correspondence:** Matteo Willeit (willeit@pik-potsdam.de)

**Abstract.** The carbon cycle component of the newly developed Earth System Model of intermediate complexity CLIMBER-X is presented. The model represents the cycling of carbon through atmosphere, vegetation, soils, seawater and marine sediments. Exchanges of carbon with geological reservoirs occur through sediment burial, rock weathering and volcanic degassing. The state-of-the-art HAMOCC6 model is employed to simulate ocean biogeochemistry and marine sediments processes. The land model PALADYN simulates the processes related to vegetation and soil carbon dynamics, including permafrost and peatlands. The dust cycle in the model allows for an interactive determination of the input of the micro-nutrient iron into the ocean. A rock weathering scheme is implemented into the model, with the weathering rate depending on lithology, runoff and soil temperature. CLIMBER-X includes a simple representation of the methane cycle, with explicitly modelled natural emissions from land and the assumption of a constant residence time of $CH_4$ in the atmosphere. Carbon isotopes $^{13}C$ and $^{14}C$ are tracked through all model compartments and provide a useful diagnostic for model-data comparison.

A comprehensive evaluation of the model performance for present–day and the historical period shows that CLIMBER-X is capable of realistically reproducing the historical evolution of atmospheric $CO_2$ and $CH_4$, but also the spatial distribution of carbon on land and the 3D structure of biogeochemical ocean tracers. The analysis of model performance is complemented by an assessment of carbon cycle feedbacks and model sensitivities compared to state-of-the-art CMIP6 models.

Enabling interactive carbon cycle in CLIMBER-X results in a relatively minor slow-down of model computational performance by $\sim20\%$, compared to a throughput of $\sim10,000$ simulation years per day on a single node with 16 CPUs on a high performance computer in a climate–only model setup. CLIMBER-X is therefore well suited to investigate the feedbacks between climate and the carbon cycle on temporal scales ranging from decades to $>100,000$ years.

# 1  Introduction

Atmospheric $CO_2$ exerts a profound control on the state of the Earth system. Although it is present only in tiny concentrations in the present-day atmosphere, by absorbing radiation in the longwave spectral range it has a substantial effect on the energy balance of the Earth. In the present day atmosphere, $CO_2$ is the second most important greenhouse gas, after water vapor. $CO_2$ is also a fundamental molecule for life on Earth, as it serves as 'food' in the photosynthesis process. The atmospheric $CO_2$ concentration is hence a main control on the growth rate of plants on land.

From ice core data it is well known that atmospheric $CO_2$ concentrations showed pronounced variations over the last million years (e.g. Petit et al., 1999; Augustin et al., 2004) that played an important role for the climate evolution over the Pleistocene (last ~2.6 million years) by amplifiying the variations associated with glacial-interglacial cycles (e.g. Ganopolski and Calov, 2011; Abe-Ouchi et al., 2013). Furthermore, on even longer time scales, a secular decrease in $CO_2$ is thought to have been the main driver of the gradual cooling over the Cenozoic (last 66 million years) (e.g. Raymo and Ruddiman, 1992).

Over the last few centuries, human activities have strongly disrupted the natural $CO_2$ balance, by directly emitting $CO_2$ from fossil sources into the atmosphere. The resulting increase in atmospheric $CO_2$ has been the main factor for the observed rapid climate warming since the preindustrial period (e.g. Gulev et al., 2021).

Modelling the atmospheric $CO_2$ concentration is thus fundamental both for understanding past climate changes and for predicting the future evolution of the Earth system under different anthropogenic emission scenarios. However, it is far from trivial, because atmospheric $CO_2$ is the result of complex biogeochemical processes on land, in the ocean, in marine sediments and in the lithosphere. Additionally, because of the long time scales involved in some of the carbon cycle processes, the interactive simulation of atmospheric $CO_2$ has been, and still is, a challenge for state-of-the-art Earth system models. Fast Earth system models of intermediate complexity have therefore been extensively employed for investigating carbon cycle-climate feedbacks (e.g. Bern3D (Müller et al., 2008; Tschumi et al., 2011; Stocker et al., 2013), cGENIE (Ridgwell et al., 2007; Cao et al., 2009), CLIMBER-2 (Brovkin et al., 2002, 2007, 2012), iLOVECLIM (Bouttes et al., 2015), LOVECLIM (Goosse et al., 2010) and Uvic (Eby et al., 2009; Zickfeld et al., 2011; Mengis et al., 2020)). Among these, CLIMBER-2 has successfully reproduced glacial-interglacial variations in $CO_2$ (Ganopolski and Brovkin, 2017; Willeit et al., 2019), but some of the processes involved remain uncertain. CLIMBER-X builds on the past experience in modelling the global carbon cycle with CLIMBER-2, but adds an improved and more detailed representation of carbon cycle processes both on land and in the ocean. Improvements include a generally higher spatial resolution, a 3D ocean model, a state-of-the-art ocean biogeochemistry and marine sediment model, a more comprehensive description of vegetation and soil carbon processes, including permafrost and peatlands, and a new chemical weathering scheme.

In the following, the biogeochemistry components of CLIMBER-X are presented. The climate core of CLIMBER-X is described in detail in Willeit et al. (2022).

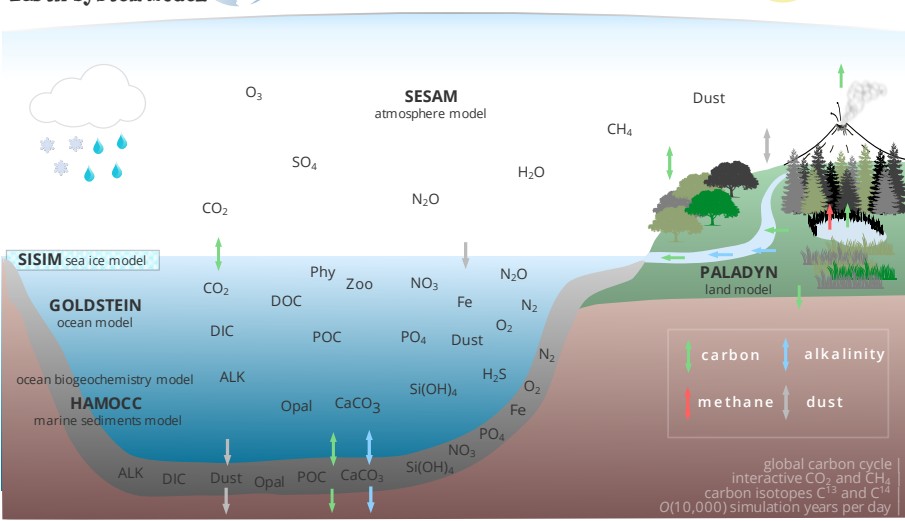

**Figure 1.** Schematic illustration of the natural biogeochemical cycles in the CLIMBER-X model.

## 2 Model description

CLIMBER-X represents the cycling of carbon through atmosphere, vegetation, soils, seawater and marine sediments. Through sediment burial, chemical weathering of rocks and volcanic degassing, carbon is also exchanged with geological reservoirs. A schematic illustration of the carbon cycle in the model is shown in Fig. 1. The carbon cycle component of CLIMBER-X consists of the ocean biogeochemistry and marine sediment models from HAMOCC6 (Maier-Reimer and Hasselmann, 1987; Ilyina et al., 2013; Heinze et al., 1999; Mauritsen et al., 2019) and the land model PALADYN (Willeit and Ganopolski, 2016), which includes dynamic vegetation, a soil carbon model and the weathering model of Hartmann (2009) and Börker et al. (2020). The atmospheric $CO_2$ concentration is determined interactively by the exchange of carbon between atmosphere, seawater, land and lithosphere. The model includes a representation of the dust cycle, with simulated dust deposition determining the input of the micro-nutrient iron into the ocean. CLIMBER-X also includes a simple representation of the methane cycle, with explicitly modelled natural emissions from land and the assumption of a constant residence time of $CH_4$ in the atmosphere. The model is enabled with the carbon isotopes $^{13}C$ and $^{14}C$, which are tracked through all model compartments.

The different model components are described in more detail in the following sections.

### 2.1 Ocean biogeochemistry and marine sediments: HAMOCC

HAMOCC (Maier-Reimer and Hasselmann, 1987; Maier-Reimer et al., 1993; Ilyina et al., 2013) is a state-of-the-art ocean biogeochemistry model, which is part of the MPI-ESM, the Earth system model of the Max Planck Institute for Meteorology

(MPI). The latest version (Mauritsen et al., 2019), which is the version employed by the MPI in the Coupled Model Intercomparison Project Phase 6 (CMIP6), has been the starting point for the implementation of the model into CLIMBER-X. As a first step, the original HAMOCC6 code has been adapted to the CLIMBER-X structure. Notably, for easier parallelisation, it has been transformed from a 3D model into a 1D vertical column model in which each water column is independent from the others. This is possible because the biogeochemical processes in the model are restricted to local vertical interactions. The different columns are interacting only through horizontal advection by ocean currents, which takes place in the ocean model.

HAMOCC represents the biogeochemical processes in the water column, the sediments, and at the air-sea interface. Marine biology dynamics is based on an extended NPZD (nutrients, phytoplankton, zooplankton, and detritus) approach (Six and Maier-Reimer, 1996). The carbonate chemistry in the model follows the latest OMIP protocol (Orr et al., 2017), which uses the robust and safe pH calculation routines from SolveSAPHE-r1 (Munhoven, 2013). In the water column, the following biogeochemical tracers are simulated: dissolved inorganic carbon (DIC), total alkalinity (TA), phosphate ($PO_4$), nitrate ($NO_3$), nitrous oxide ($N_2O$), dissolved nitrogen gas ($N_2$), silicate ($SiO_2$), dissolved bioavailable iron (Fe), dissolved oxygen ($O_2$), phytoplankton (Phy), zooplankton (Zoo), dissolved organic matter (DOC), particulate organic matter (POC), opal shells, calcium carbonate shells ($CaCO_3$), terrigenous material (dust) and hydrogen sulfide ($H_2S$). The composition of organic material follows a constant Redfield ratio (C:N:P:$O_2$ = 122:16:1:-172) after Takahashi et al. (1985), and for the micronutrient iron (Fe:C = $4\times10^{-6}$:1).

The marine sediment module, which is part of HAMOCC, is based on Heinze et al. (1999). It essentially simulates the same processes between dissolved tracers (DIC, TA, $PO_4$, $NO_3$, $O_2$, Fe, $SiO_2$, $H_2S$ and $N_2$) in pore water and solid sediment constituents (POC, opal, $CaCO_3$ and dust) as in the water column. Pore water tracers are exchanged with the overlying water column via diffusion. Sedimentation fluxes of POC, $CaCO_3$, opal and dust are added to the solid components of the sediment. Accumulation of solid sediment material will lead to active sediment layer content being shifted to the burial layer and back if boundary conditions changes lead to chemical erosion of previously buried sediment.

Next we describe the changes introduced into HAMOCC as part of its implementation into CLIMBER-X.

$N_2$ fixation is represented by a diagnostic formulation, whereby the nitrate influx into the surface layer is a function of the nitrate deficit relative to phosphate, multiplied by a constant fixation rate (Ilyina et al., 2013). Prognostic $N_2$ fixers have recently been included in HAMOCC (Paulsen et al., 2017), based on the physiological characteristics of the cyanobacterium Trichodesmium. However, for simplicity and because uncertainties in nitrogen fixation remain large (e.g. Zehr and Capone, 2020), in CLIMBER-X cyanobacteria are disabled by default.

Following Heinemann et al. (2019), we have implemented a representation of aggregates in the model. Particulate organic carbon is assumed to form aggregates with the denser calcite and opal built during phytoplankton and zooplankton growth, as well as with dust particles. The sinking speed of these aggregates depends on their excess density (Gehlen et al., 2006; Heinemann et al., 2019). Note that this approach neglects the effects of, e.g., aggregate size distribution and porosity on the sinking speed (Maerz et al., 2020), and it does not, like other numerically more expensive schemes (e.g., Kriest and Evans, 2000) explicitly resolve the biological and physical aggregation and disaggregation processes. The introduction of the ballasting scheme required a retuning of the dissolution rates of calcite and opal as shown in Table 1.

Following recent evidence that the remineralisation of organic carbon depends on temperature (e.g. Laufkötter et al., 2017), we have introduced a Q10 temperature dependence for the remineralisation of POC and DOC (Segschneider and Bendtsen, 2013; Crichton et al., 2021), with a default Q10 value of 2. The complete set of remineralisation parameters is listed in Table 1.

In the original HAMOCC, iron complexation by organic substances is assumed when the iron concentration exceeds a given threshold and dissolved iron is then removed from the water column at a fixed rate. In CLIMBER-X, we explicitly model iron complexation, differentiating between free and complexed iron forms following Archer and Johnson (2000) and Parekh et al. (2004). The complexed iron is associated with an organic ligand and only the free iron is available for scavenging. The ligand concentration is assumed to be constant at $1 \, \mathrm{nmol kg^{-1}}$ with a ligand stability constant of $1 \times 10^{11} \, \mathrm{kg mol^{-1}}$. The speciation of iron is then determined by equilibrium kinetics. The scavenging rate of free iron is a combination of a minimum scavenging rate and a scavenging rate that is proportional to the POC, calcite and opal concentrations following Aumont et al. (2015) and Hauck et al. (2013). Compared to HAMOCC we have also increased the stochiometric iron ratio in organic compounds from Fe:C = $3 \times 10^{-6}$:1 to Fe:C = $4 \times 10^{-6}$:1. The parameters related to the iron cycle are also reported in Table 1.

The carbon 13 isotope has been recently implemented in HAMOCC by Liu et al. (2021). In CLIMBER-X we extended this approach to also include radiocarbon.

Since the ocean model in CLIMBER-X is a rigid lid model, following the OMIP protocol (Orr et al., 2017), we explicitly take into account the local concentration-dilution effect of the net surface freshwater flux, which changes surface DIC concentration and alkalinity. Two options are available in the model to implement the dilution effect on DIC and alkalinity. The first one ensures that the net global surface tracer flux is zero by applying deviations from the global average freshwater flux to the global average surface tracer concentration. The second (default) option applies the actual local surface freshwater flux to compute a new virtual top ocean layer thickness and then dilutes the tracers accordingly. In this case, the conservation of tracer inventories is ensured by compensating disbalances over the global ocean. Additionally, during times when ocean volume is changing because of buildup or melt of land ice, concentrations of all tracers are globally adjusted while conserving tracer inventories. This is a reasonable simplification, considering that land ice volume changes occur on multi-millenial time scales, over which the ocean can be considered well mixed.

Based on scale analysis, we have excluded fast sinking tracers ($CaCO_3$, opal, POC and dust) from advection, as these particles have sinking speeds which are large enough so that vertical transfer between different grid cells is more rapid than horizontal transfer by advection would be, considering the relatively coarse resolution of the ocean model. Following a similar line of thought also short-lived tracers like phytoplankton and zooplankton are excluded from oceanic transport. However, convection and wind–driven surface vertical mixing are applied to all biogeochemical tracers.

In CLIMBER-X, HAMOCC is integrated with a time step of one day, which is also the time step of the physical ocean model.

## 2.2 Land carbon cycle: PALADYN

PALADYN is a comprehensive land surface-vegetation-carbon cycle model designed specifically for the use in CLIMBER-X (Willeit and Ganopolski, 2016). It includes a detailed representation of the land carbon cycle. Photosynthesis is computed

**Table 1.** Modified HAMOCC parameters used in CLIMBER-X compared to HAMOCC6 (i.e. Table 2 in Ilyina et al. (2013)).

| Parameter | Description | CLIMBER-X | HAMOCC6 | Unit |
|---|---|---|---|---|
| *Nutrients* | | | | |
| $\mu_{\mathrm{cyan}}$ | $N_2$ fixation rate | 0.0025 | 0.005 | $d^{-1}$ |
| $\lambda_{\mathrm{det}}^{\mathrm{ref}}$ | POC remineralisation rate at temperature $T_{\mathrm{ref}}$ | 0.05 | 0.025 | $d^{-1}$ |
| $\lambda_{\mathrm{N}}^{\mathrm{ref}}$ | denitrification rate at temperature $T_{\mathrm{ref}}$ | 0.15 | 0.07 | $d^{-1}$ |
| $\lambda_{\mathrm{S}}^{\mathrm{ref}}$ | sulfate reduction rate at temperature $T_{\mathrm{ref}}$ | 0.005 | 0.005 | $d^{-1}$ |
| $Q_{10}$ | Q10 for temperature dependence of remineralisation rate | 2 | 1 | |
| $T_{\mathrm{ref}}$ | reference temperature for remineralisation rate | 20 | | $^{\circ}\mathrm{C}$ |
| *Iron cycle* | | | | |
| $f_{\mathrm{Fe}}^{\mathrm{dust}}$ | fraction of iron mass in dust | 0.025 | 0.035 | $\mathrm{kgkg}^{-1}$ |
| $d_{\mathrm{Fe}}$ | iron solubility in surface water | 0.01 | 0.01 | |
| $R_{\mathrm{Fe:C}}$ | stochiometric iron ratio in organic compounds | $4\times10^{-6}$ | $3\times10^{-6}$ | $\mathrm{molFemolC}^{-1}$ |
| $L$ | total ligand concentration | $1\times10^{-9}$ | | $\mathrm{kmolm}^{-3}$ |
| $k_{\mathrm{L}}$ | ligand stability constant | $1\times10^{11}$ | | $\mathrm{m^3kmol}^{-1}$ |
| $k_{\mathrm{scav}}^{\mathrm{min}}$ | minimum free Fe scavenging rate | $3\times10^{-5}$ | | $d^{-1}$ |
| $k_{\mathrm{scav}}^{\mathrm{POC}}$ | slope of free Fe scavenging rate by POC | 0.002 | | $\left(\mathrm{mmolCm}^{-3}\right)^{-1}\mathrm{d}^{-1}$ |
| $k_{\mathrm{scav}}^{\mathrm{shells}}$ | slope of free Fe scavenging rate by shells | 0.002 | | $\left(\mathrm{mmol(C/Si)m}^{-3}\right)^{-1}\mathrm{d}^{-1}$ |
| *Shell material* | | | | |
| $K_{\mathrm{SiO}}$ | half-saturation constant for $Si(OH)_4$ uptake | $5\times10^{-6}$ | $1\times10^{-6}$ | $\mathrm{kmolSim}^{-3}$ |
| $R_{\mathrm{Ca:P}}$ | $CaCO_3$:P uptake ratio | 10 | 20 | $\mathrm{molCmolP}^{-1}$ |
| $R_{\mathrm{Si:P}}$ | opal:P uptake ratio | 50 | 25 | $\mathrm{molSimolP}^{-1}$ |
| $\lambda_{\mathrm{calc}}$ | dissolution rate of calcite shells | 0.3 | 0.075 | $d^{-1}$ |
| $\lambda_{\mathrm{opal}}$ | dissolution rate of opal shells | 0.0025 | 0.01 | $d^{-1}$ |
| *Sediments* | | | | |
| $\lambda_{\mathrm{det}}^{\mathrm{sed,ref}}$ | sediment POC remineralisation rate at temperature $T_{\mathrm{ref}}^{\mathrm{sed}}$ | 0.025 | 0.01 | $\left(\mathrm{kmolO_2m}^{-3}\right)^{-1}\mathrm{d}^{-1}$ |
| $\lambda_{\mathrm{N}}^{\mathrm{sed,ref}}$ | sediment denitrification rate at temperature $T_{\mathrm{ref}}^{\mathrm{sed}}$ | 0.1 | 0.01 | $d^{-1}$ |
| $\lambda_{\mathrm{S}}^{\mathrm{sed,ref}}$ | sediment sulfate reduction rate at temperature $T_{\mathrm{ref}}^{\mathrm{sed}}$ | $2.5\times10^{-5}$ | $2.5\times10^{-5}$ | $d^{-1}$ |
| $Q_{10}^{\mathrm{sed}}$ | Q10 for temperature dependence of remineralisation rate | 2 | 1 | |
| $T_{\mathrm{ref}}^{\mathrm{sed}}$ | reference temperature for remineralisation rate in sediments | 5 | | $^{\circ}\mathrm{C}$ |
| $\lambda_{\mathrm{calc}}^{\mathrm{sed}}$ | sediment dissolution rate constant of $CaCO_3$ | 0.02 | 0.0086 | $\left(\mathrm{kmolCaCO_3m}^{-3}\right)^{-1}\mathrm{d}^{-1}$ |
| $\lambda_{\mathrm{opal}}^{\mathrm{sed}}$ | sediment dissolution rate constant of opal | 0.005 | 0.0026 | $\left(\mathrm{kmolSi(OH)_4m}^{-3}\right)^{-1}\mathrm{d}^{-1}$ |

following the Farquhar model (Farquhar et al., 1980; Collatz et al., 1991) and depends on absorbed shortwave radiation, air temperature, vapor pressure deficit between leaf and ambient air, atmospheric $CO_2$ and soil moisture. Carbon assimilation by vegetation is coupled to the transpiration of water through stomatal conductance. The model includes a dynamic vegetation module with 5 plant functional types (PFTs) competing for the gridcell share based on their respective net primary productivity. The model distinguishes between mineral soil carbon, peat carbon, buried carbon and shelf carbon. Each soil carbon 'type'

has its own soil carbon pools generally represented by a litter, a fast and a slow carbon pool in each of the five soil layers. Carbon can be redistributed between the layers by vertical diffusion. For the vegetated macro surface type, decomposition is a function of soil temperature and soil moisture. Carbon in permanently frozen layers is assigned a long turnover time which effectively locks carbon in permafrost. Carbon buried below ice sheets and on ocean shelfs is treated separately. The land model also includes a dynamic peat module. PALADYN includes carbon isotopes $^{13}$C and $^{14}$C, which are tracked through all carbon

pools in vegetation and soil. Isotopic discrimination is modelled only during the photosynthetic process. A simple methane module is implemented to represent methane emissions from anaerobic carbon decomposition in wetlands and peatlands. The integration of PALADYN into the coupled CLIMBER-X framework and subsequent sensitivity analyses of the land carbon cycle feedbacks, which were not performed with the offline PALADYN setup in Willeit and Ganopolski (2016), highlighted the need to improve certain aspects of the model. These improvements are described next.

We have updated the parameterisation of the roughness length for heat and moisture. Originally, it was simply taken to be proportional to the roughness length for momentum, but there is ample evidence from observations that the roughness length for scalars can be orders of magnitude lower than that for momentum when the surface roughness is large (e.g. Zilitinkevich, 1995; Chen and Zhang, 2009; Yang et al., 2008; Zheng et al., 2012). We have therefore implemented the parameterisation from Zilitinkevich (1995), which includes a dependence of the surface roughness length for heat and moisture on the roughness

Reynolds number. With this new parameterisation the exchange coefficient for the turbulent surface fluxes shows a much weaker dependence on the roughness of the surface, which has an impact on the vegetation feedback.

We have introduced a topographic erodibility factor for dust emissions following Ginoux et al. (2001). It assumes that a basin with pronounced topographic variations contains large amount of sediments which have accumulated in the valleys and depressions and which can easily be mobilized by wind. The following topographic factor is then applied to scale dust

emissions:

$$f_{\text{topo}} = \left( \frac{\max(0, z_{\max} - z)}{z_{\max} - z_{\min}} \right)^5, \tag{1}$$

where $z$ is the grid-cell mean elevation, and $z_{\max}$ and $z_{\min}$ are the maximum and minimum surface elevations computed from the high-resolution topography in the surrounding $15 \times 15°$. The exponent 5 is taken from Zender et al. (2003).

The rubisco-limited photosynthesis rate in the version of PALADYN model described in Willeit and Ganopolski (2016)

was based on the 'strong optimality' hypothesis of Haxeltine and Prentice (1996), which assumes that rubisco activity and the nitrogen content of leaves vary with canopy position and seasonally so as to maximize net assimilation at the leaf level (Schaphoff et al., 2018). However, we found that this formulation led to a relatively small increase in gross primary production over the historical period, which resulted into an overestimation of atmospheric $CO_2$ in coupled historical simulations. We

therefore introduced a new formulation for the maximum rubisco capacity, with dependencies on PFT-specific, constant foliage nitrogen concentration, specific leaf area and leaf temperature following Thornton and Zimmermann (2007) as implemented in CLM4.5 (Oleson et al., 2010).

In the original PALADYN formulation, the internal leaf $CO_2$ concentration used for photosynthesis was computed based on the Cowan–Farquhar optimality hypothesis (Medlyn et al., 2011). In the new model version, for C3 plants, we have implemented an alternative scheme following the more general least-cost optimality model (Prentice et al., 2014; Lavergne et al., 2019) with the moisture dependence proposed by Lavergne et al. (2020).

In the isotopic discrimination during photosynthesis ($\Delta$) we included an explicit fractionation term for photorespiration as recommended by several recent studies (Ubierna and Farquhar, 2014; Schubert and Jahren, 2018; Lavergne et al., 2019):

$$\Delta = 4.4\frac{c_a - c_i}{c_a} + 27 \cdot \frac{c_i}{c_a} - 12\frac{\Gamma_\star}{p_a}, \tag{2}$$

where $c_a$ and $c_i$ are the ambient and leaf internal $CO_2$ concentrations, $p_a$ is the ambient partial pressure of $CO_2$ and $\Gamma_\star$ is the $CO_2$ compensation point.

For the distinction between evergreen and summergreen trees, in addition to a threshold on coldest month temperature we have introduced a PFT-specific threshold on the growing degree days above $5°C$, which is set to 600 for needleleaf trees and 900 for shrubs following Sitch et al. (2003).

In the dynamic vegetation model a parameter ($\lambda$) is used to partition the net primary production (NPP) between local growth of existing vegetation and lateral expansion ('spreading') of vegetation coverage within the grid cell, with all of the NPP being used for growth for small leaf area index (LAI) values, and all the NPP being used for 'spreading' for large LAI values. $\lambda$ is assumed to be a piecewise linear function of the leaf area index between a minimum and maxium LAI. For small leaf area indices, all of the NPP is used for local growth ($\lambda = 0$); for LAI above a critical value $LAI_{min}$, a fraction ($\lambda > 0$) is used for 'spreading':

$$\lambda = \frac{LAI - LAI_{min}}{LAI_{max} - LAI_{min}}. \tag{3}$$

However, since the simulated leaf area index depends strongly on NPP, which in turn has a pronounced dependence on atmospheric $CO_2$, this formulation results in a strong dependence of $\lambda$ on $CO_2$, with an increasingly larger fraction of NPP being used for 'spreading' as $CO_2$ increases. We have therefore implemented a $CO_2$ dependence in the maximum leaf area index to reduce this effect:

$$LAI_{max} = LAI_{max}^{ref} \cdot \left(1 + 0.5 \cdot \log\left(CO_2/CO_2^{ref}\right)\right). \tag{4}$$

The fraction of decomposed litter respired directly as $CO_2$ to the atmosphere has been reduced from 0.7 to 0.6 and the fraction of decomposed litter transferred to the slow soil carbon pool has been doubled from 0.015 to 0.03. Together these changes result in more carbon accumulating into the soil.

A simple representation of land use change has been introduced into the model following Burton et al. (2019) as described in Willeit et al. (2022). A fraction of each grid cell is prescribed as being used for agriculture and land use is then represented as a

limitation to the space available for the woody PFTs to expand into. When forests and shrubs are affected by land use change, an additional disturbance rate of $1\,\mathrm{yr}^{-1}$ is prescribed on top of the standard background disturbance, leading to vegetation dying. The resulting dead vegetation carbon is then added as litter to the soil carbon pools, and a large part will be respired directly to the atmosphere within a few years. Storage of wood from deforestation into products such as paper or wood for construction is not accounted for in the model and soil carbon is assumed to not be directly affected by land use practices. Following deforestation, the model will grow C3 or C4 grasses, depending on climate conditions.

The partitioning of the soil carbon decomposed under anaerobic conditions into $CO_2$ and $CH_4$ used a prescribed constant ratio in Willeit and Ganopolski (2016). We modified this by making the fraction released as $CH_4$ dependent on temperature with a Q10 of 1.8, following Riley et al. (2011) and Kleinen et al. (2020).

We implemented a chemical weathering model to compute the riverine fluxes of bicarbonate ions ($HCO_3^-$), (and therefore dissolved inorganic carbon and alkalinity) to the ocean and the consumption of atmospheric $CO_2$. The weathering rate depends on the lithology and on the climate variables temperature and runoff. The lithological map of Hartmann and Moosdorf (2012) distinguishing 16 different lithologies is used to describe the spatial distribution of rocks. The parameters for the chemical weathering equations for all lithologies, except for carbonate sedimentary rocks and loess, are based on a spatially explicit runoff-dependent model of chemical weathering, which was calibrated for 381 catchments in Japan (Hartmann, 2009), with the additional temperature dependence of Hartmann et al. (2014). The effect of soil shielding on the weathering rate suggested by Hartmann et al. (2014) has not been considered since information on soil shielding is not readily available for periods beyond the recent past. For carbonate sedimentary rocks, the weathering rate follows the approach of Amiotte Suchet and Probst (1995) with a dependence on runoff. Alternatively, the temperature dependent formulation of Romero-Mujalli et al. (2019) is available for use in the model. The weathering rate for loess sediments depends on runoff following Börker et al. (2020). The global distribution of loess cover for present-day and for the last glacial maximum, as well as the lithologies of the continental shelves that were exposed at the last glacial maximum, are taken from Börker et al. (2020). The weathering fluxes are transferred from the land to the ocean in the same way as water runoff, following the runoff routing scheme.

The carbon isotopes fluxes from chemical weathering are computed assuming a $\delta^{13}C$ of $1.8\,\mathrm{permil}$ for carbon originating from carbonate minerals (Derry and France-Lanord, 1996).

Equations describing silicate and phosphorus weathering fluxes are also available as part of the weathering model. However, silicate and phosphorus riverine fluxes are not considered in the default model setup, as they would result in further complications related to the conservation of nutrients in the ocean. Instead, as discussed in Sect. 2.1, the silicate and phosphorus budgets are closed by assuming that the sediment burial flux is returned as input at the ocean surface.

## 2.3 Atmospheric $CO_2$

The atmospheric $CO_2$ concentration in CLIMBER-X is a globally uniform value. It can either be prescribed (as constant or time-dependent) or interactively computed by the model from the following prognostic equation for the total carbon content

stored as $CO_2$ in the atmosphere ($C_{atm}$):

$$\frac{dC_{atm}}{dt} = F_{ocn} + F_{lnd} + F_{anth} - F_{weath} + F_{volc} + F_{CH_4ox}. \tag{5}$$

The source and sink terms on the right hand side represent, from left to right, the net sea-air carbon flux, the global net land to atmosphere carbon flux, the anthropogenic carbon emissions (excluding land-use change), the $CO_2$ consumption by silicate and carbonate weathering, the volcanic degassing flux and the $CO_2$ flux from the oxidation of atmospheric methane originating from non-agricultural sources. The $CO_2$ consumption by weathering is computed assuming that all carbon in the $HCO_3^-$ originating from the weathering of silicate rocks ($F_{HCO_3^-}^{sil}$) comes from the atmosphere, while only half of the carbon in the $HCO_3^-$ originating from the weathering of carbonate rocks and sediments ($F_{HCO_3^-}^{carb}$) comes from the atmosphere:

$$F_{weath} = F_{HCO_3^-}^{sil} + 0.5 \cdot F_{HCO_3^-}^{carb}. \tag{6}$$

The constant volcanic degassing rate is set to half the silicate weathering rate (e.g. Munhoven and François, 1994) as determined by an equilibrium spinup simulation:

$$F_{volc} = 0.5 \cdot F_{HCO_3^-}^{sil}. \tag{7}$$

The flux from the oxidation of methane, $F_{CH_4ox}$, is computed by the $CH_4$ model as described in Sect. 2.4 below. The atmospheric $CO_2$ concentration is then computed from $C_{atm}$ using a conversion factor of $2.12\,\mathrm{PgCppm}^{-1}$ (Denman et al., 2007).

Equations similar to eq. 5 are used also for the carbon isotopes $^{13}C$ and $^{14}C$. The prognostic equation for the stable isotope $^{13}C$ in atmospheric $CO_2$ is:

$$\frac{d^{13}C_{atm}}{dt} = F_{ocn}^{13} + F_{lnd}^{13} + F_{anth}^{13} - F_{weath}^{13} + F_{volc}^{13}. \tag{8}$$

The $^{13}C$ fluxes from land and ocean are explicitly computed by the land and ocean carbon cycle models as described in detail in Willeit and Ganopolski (2016) and Liu et al. (2021). The $\delta^{13}C$ of anthropogenic carbon emissions is prescribed as time-dependent from historical data of Andres et al. (2016) and the $^{13}C$ flux from $CO_2$ consumption by weathering, assuming no fractionation, is simply computed as:

$$F_{weath}^{13} = F_{weath}\frac{^{13}C_{atm}}{C_{atm}}. \tag{9}$$

The $^{13}C$ of volcanic degassing is computed assuming a $\delta^{13}C$ of -5 permil.

The prognostic equation for radiocarbon $^{14}C$ in atmospheric $CO_2$ reads:

$$\frac{d^{14}C_{atm}}{dt} = F_{ocn}^{14} + F_{lnd}^{14} - F_{weath}^{14} + F_{prod}^{14} - \frac{^{14}C_{atm}}{\tau_{14C}}. \tag{10}$$

Carbon sources originating from geological reservoirs, i.e. volcanic degassing, are assumed to contain no radiocarbon. Similarly, radiocarbon is assumed to be absent in anthropogenic carbon emissions from fossil fuel burning, because the age of fossils far exceeds the half-life of $^{14}C$. The production rate of radiocarbon in the atmosphere ($F_{prod}^{14}$) is prescribed in the model and the radiocarbon decay time is $\tau_{14C} = 8267$ yr.

## 2.4 Atmospheric $CH_4$

Similarly to $CO_2$, atmospheric $CH_4$ is also considered to be well-mixed in the atmosphere and is therefore represented as a globally uniform value. The atmospheric $CH_4$ concentration can be prescribed, or it can be interactively computed by the model from:

$$\frac{d\text{CH}_4}{dt} = F_{\text{lnd}}^{emis} + F_{\text{anth}}^{emis} - \frac{\text{CH}_4}{\tau_{\text{CH}_4}}. \tag{11}$$

Methane sources include natural emissions from wetlands and peatlands ($F_{\text{lnd}}^{emis}$), which are explicitly simulated by the model as originating from anaerobic decomposition processes of carbon in soils (Willeit and Ganopolski, 2016). Other natural sources of methane are generally smaller (e.g. Saunois et al., 2020; Kleinen et al., 2020) and are neglected here for simplicity. Anthropogenic methane emissions ($F_{\text{anth}}^{emis}$) are prescribed in the model. The sink of methane from oxidation in the atmosphere is computed using a constant residence time of $CH_4$, $\tau_{\text{CH}_4} = 9.5\,\text{years}$, which is a reasonable first approximation at least for climate conditions ranging between the last glacial maximum and present-day (Kleinen et al., 2020; Levine et al., 2011; Hopcroft et al., 2017).

## 3 Closed and open carbon cycle model configurations and model spinup

Two different configurations of the carbon cycle model are available and can be chosen according to the specific needs.

The first (and simplest) setup consists of ocean, land and atmosphere carbon cycle components only. In this setup marine sediments are disabled and particulate fluxes that reach the ocean floor are completely remineralised/dissolved in the bottom ocean grid cell. Rock weathering from land is also switched off, so that the carbon exchange between ocean, land and atmosphere occurs only through air-sea fluxes and through land-atmosphere exchanges. In this setup the carbon system is closed, in the sense that there are no natural sources and sinks from and to geological reservoirs. As a response to an external climate perturbation, carbon is then simply redistributed between atmosphere, ocean and land, with the total carbon in the system being conserved. This setup is equivalent to what is used in many state-of-the-art Earth System models for climate change projections on centennial time scales (e.g. Séférian et al., 2020). The model spinup for this simple setup is straightforward and requires only to run the model to steady state with a prescribed atmospheric $CO_2$ concentration for $\approx 10,000$ years. The slowest time scale in this setup is given by the slow decomposition rate of organic carbon in frozen soils, which is limited to a maximum value set by default to 5000 years. The initial state for the spinup run is given by observed present–day 3D concentrations of different tracers in the ocean (Lauvset et al., 2016; Olsen et al., 2016; Garcia et al., 2013b), while the land surface is assumed to be covered by bare soil and with no carbon stored on land.

The closed carbon cycle setup is applicable to simulations of up to $1000\,\text{yr}$. On longer time scales, sediment and weathering processes become important and need to be accounted for when performing long-term transient simulations with interactive $CO_2$. Although it is unlikely that in reality the slow carbon cycle processes related to marine sediments, peatlands and permafrost carbon are in equilibrium at any specific point in time, for practical reasons we assume that such an equilibrium is a reasonable first approximation. Assuming that the preindustrial is an equilibrium state of the climate-carbon cycle system

allows to run perturbation experiments with the interactive carbon cycle without having to deal with possible long-term drifts in atmospheric $CO_2$. However, the long time scale of $\sim$100,000 years involved in ocean sediment processes represents a challenge in running the model into equilibrium, even for a high-throughput model like CLIMBER-X. We therefore implemented a scheme to run the physical ocean and ocean biogeochemistry models in an offline setup with prescribed climatological daily input fields at the ocean surface. This setup results in a speedup of a factor >2 relative to running the fully coupled climate-carbon cycle model, meaning that ocean carbon cycle and marine sediments can be run into equilibrium in about a week of computing time on a high performance computer. In detail, the spinup procedure of the full carbon cycle configuration comprises two different stages. Atmospheric $CO_2$ is prescribed to a constant value throughout the process, at 280 ppm for the pre–industrial case. The first stage aims at spinning up the sediment model. For this purpose the full carbon-cycle climate model is run for 5,000 years and every 300 years the sediment model is run offline for 1000 years. During this stage all net fluxes into the sediments are compensated for and returned as inputs at the ocean surface in order to approximately conserve water column tracer inventories while the sediments are filling up. In the second stage we switch to simulated DIC and alkalinity weathering fluxes from land and at the same time also switch to the more efficient offline ocean–biogeochemistry setup described above and run the model until an approximate equilibrium is reached after $\sim$100,000 years (Fig. 2). A simplification that is made in the open carbon cycle setup is that organic carbon and opal that are buried into the sediments, and are therefore effectively leaving the system, are returned in remineralized form to the surface ocean so that phosphorus and silica inventories of the ocean–sediment system are conserved throughout the simulation.

The carbon fluxes among the different model components in the open setup for equilibrium pre-industrial conditions are schematically illustrated in Fig. 3. The volcanic degassing rate is equal to half the atmospheric $CO_2$ consumption by silicate weathering, in accordance with theory (Munhoven and François, 1994). Note that not only the carbon budget of the different compartments (atmosphere, ocean, lithosphere) is well balanced, but also the ocean alkalinity budget is.

## 4   Model evaluation for historical period and present day

Here we present results from a CLIMBER-X simulation with interactive $CO_2$ and $CH_4$ in the open carbon cycle setup for the historical period (1850–2015) and provide a comprehensive evaluation of model performance against various observational datasets. The forcings for this simulation include variations in solar radiation (Matthes et al., 2017), radiative forcing of volcanic eruptions (Prather et al., 2013), globally uniform $N_2O$ concentrations from Köhler et al. (2017), globally uniform CFC11 and CFC12 concentrations from Meinshausen et al. (2016), 3D $O_3$ concentrations and 2D $SO_4^{2-}$ load from the ensemble mean of CMIP6 models and land use change (pasture and cropland fractions) from Ma et al. (2020). The model is initialized from an 80,000–year equilibrium simulation with the open carbon cycle setup for pre-industrial boundary conditions and a prescribed atmospheric $CO_2$ of 280 ppm, as described in section 3 and shown in Fig. 2.

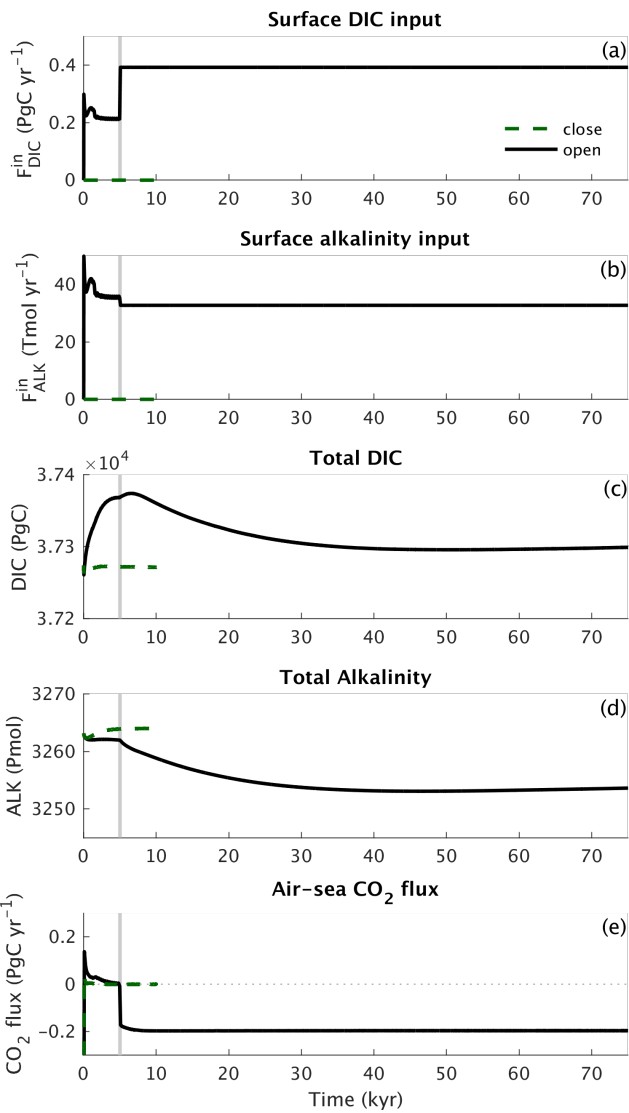

**Figure 2.** Open versus closed carbon cycle spinup for pre-industrial conditions. The figure shows surface input of (a) DIC and (b) alkalinity, the evolution of (c) DIC and (d) alkalinity inventories in the ocean and (e) the air-sea $CO_2$ flux. The grey vertical lines indicate the switch between first and second spinup phase, as described in the text.

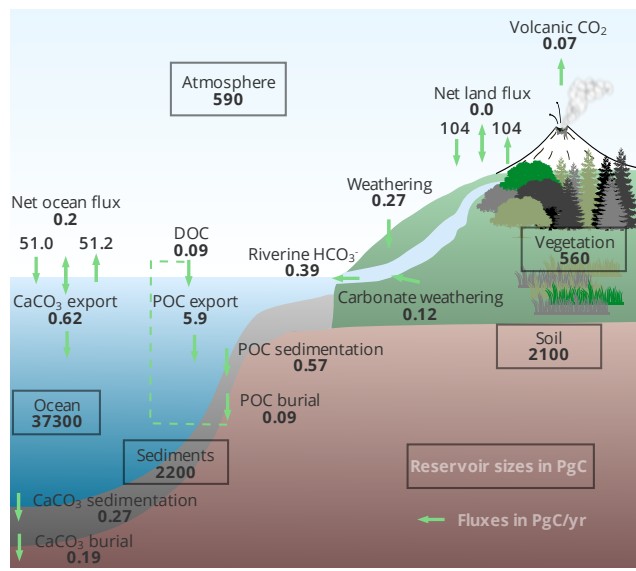

**Figure 3.** CLIMBER-X carbon fluxes and reservoirs in equilibrium with pre-industrial conditions for the open carbon cycle setup.

## 4.1 Present day

In the following, different simulated climatological characteristics are compared to observations to assess the model performance for present day. Unless stated otherwise the comparison with observations is for the time interval from 1981 to 2010. To give an overview of how CLIMBER-X compares to state-of-the-art Earth system models based on general circulation models, we also include results from model simulations from the recent coupled model intercomparison project CMIP6 (Eyring et al., 2016). The following CMIP6 models are included for ocean biogeochemistry: CESM2, IPSL-CM6A-LR, MRI-ESM2-0, MIROC-ES2L, MPI-ESM1-2-LR, UKESM1-0-LL and CanESM5. For the land carbon cycle, the following models are used for comparison: ACCESS-ESM1-5, BCC-CSM2-MR, CanESM5, CNRM-ESM2-1, GFDL-ESM4, IPSL-CM6A-LR, MIROC-ES2L, MPI-ESM1-2-LR, MRI-ESM2-0, NorESM2-LM, UKESM1-0-LL. For ocean biogeochemistry, we highlight how the model compares with results from the MPI-ESM1-2-LR employing the original marine carbon cycle model HAMOCC6.

### 4.1.1 Ocean biogeochemistry and marine sediments

An overview of simulated global variables characterising the ocean carbon cycle are presented and compared to observation-based estimates in Table 2, providing a summary of model performance for the present day.

Representing ocean ventilation time scale reasonably well is a prerequisite for simulating biogeochemical tracers in the ocean. The ocean uptake of CFCs of anthropogenic origin over the historical period is often used to probe the ventilation of the ocean on decadal time scales, while the pre-industrial radiocarbon concentration in the ocean provides information on the age distribution of the water masses in an approximate equilibrium state. We therefore start by comparing how well the model reproduces the CFC11 and radiocarbon distributions in the ocean. The inventory of CFC11 in the ocean starts to increase

**Table 2.** Global values of the main ocean biogeochemical variables for the present-day.

| | CLIMBER-X | Estimated range | Unit | Source |
|---|---|---|---|---|
| *Ocean-Atmosphere Fluxes* | | | | |
| Pre-industrial $CO_2$ flux | 0.2 | 0.2–0.6 | $PgCyr^{-1}$ | Jacobson et al. (2007);Regnier et al. (2013) |
| $N_2O$ flux | 5.0 | 1.9–9.4 | $TgNyr^{-1}$ | Buitenhuis et al. (2018) |
| *Surface Nutrients and Alkalinity* | | | | |
| Surface alkalinity | 2410 | 2355 | $mmolm^{-3}$ | GLODAPv2, (Lauvset et al., 2016; Olsen et al., 2016) |
| Surface nitrate | 6.3 | 5.2 | $mmolNm^{-3}$ | WOA 2013, Garcia et al. (2013b) |
| Surface phosphate | 0.51 | 0.53 | $mmolPm^{-3}$ | WOA 2013, Garcia et al. (2013b) |
| Surface silicate | 8.2 | 7.5 | $mmolSim^{-3}$ | WOA 2013, Garcia et al. (2013b) |
| *Primary Production* | | | | |
| Net primary production | 53 | 47-60 | $PgCyr^{-1}$ | Johnson and Bif (2021);Carr et al. (2006) |
| N-fixation | 88 | 51-200 | $TgNyr^{-1}$ | Karl et al. (2002);Großkopf et al. (2012) |
| *Export production* | | | | |
| POC export at 100 m | 5.9 | 5.8–12.9 | $PgCyr^{-1}$ | Dunne et al. (2007) |
| $CaCO_3$ export at 100 m | 0.62 | 0.38–1.8 | $PgCyr^{-1}$ | Dunne et al. (2007) |
| Opal export at 100 m | 105 | 94.5–155.5 | $TmolSiyr^{-1}$ | Tréguer and De La Rocha (2013) |
| *Sediments* | | | | |
| POC sediment deposition | 0.57 | 0.93–3.2 | $PgCyr^{-1}$ | Dunne et al. (2007) |
| $CaCO_3$ sediment deposition | 0.27 | 0.16–0.4 | $PgCyr^{-1}$ | Battaglia et al. (2016); Milliman and Droxler (1996) |
| Opal sediment deposition | 79 | 79-84 | $PgCyr^{-1}$ | Tréguer and De La Rocha (2013);Tréguer et al. (2021) |
| POC burial | 0.09 | 0.07–0.7 | $PgCyr^{-1}$ | Cartapanis et al. (2018) |
| $CaCO_3$ burial | 0.19 | 0.13–0.45 | $PgCyr^{-1}$ | Cartapanis et al. (2018) |
| Opal burial | 5.3 | 2.7–9.9 | $TmolSiyr^{-1}$ | Tréguer and De La Rocha (2013) |

after ≈1950, as a consequence of its increase in the atmosphere (Fig. 4). Estimates for CFC11 inventory in the year ≈1994 are available from models from the OCMIP model intercomparison (Dutay et al., 2002) and from direct observations (Willey et al., 2004). CLIMBER-X results are generally consistent with these estimates (Fig. 4), indicating that, at least at the global scale, the decadal ventilation time scale in CLIMBER-X is well in line with observations and other models. In terms of spatial distribution, the CFC11 uptake is overestimated in the North Pacific, the northern Indian Ocean and around Antarctica, while too small CFC11 concentrations are simulated at mid-latitudes in all basins at depths between 500 and 1000 m (Fig. 5).

The radiocarbon ventilation age in the pre-industrial gives additional insights into the ocean ventilation in quasi-equilibrium conditions, an information which is complementary to CFC11. The radiocarbon ventilation age of the deep ocean is nicely reproduced by CLIMBER-X, while radiocarbon age is systematically overestimated in the upper kilometer across all ocean basins (Fig. 6). The too old (in terms of radiocarbon age) sub-surface waters could be a result of the model not explicitly

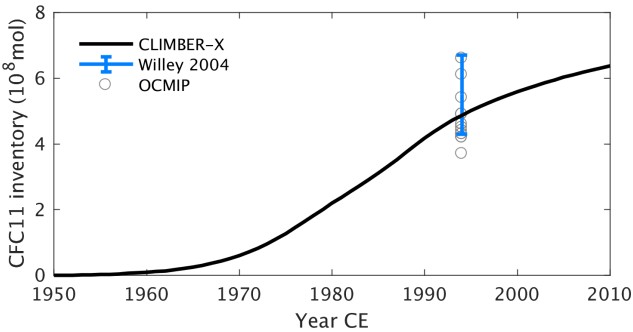

**Figure 4.** Historical global cumulative ocean uptake of CFC11 in CLIMBER-X compared to observations (Willey et al., 2004) and OCMIP models (Dutay et al., 2002).

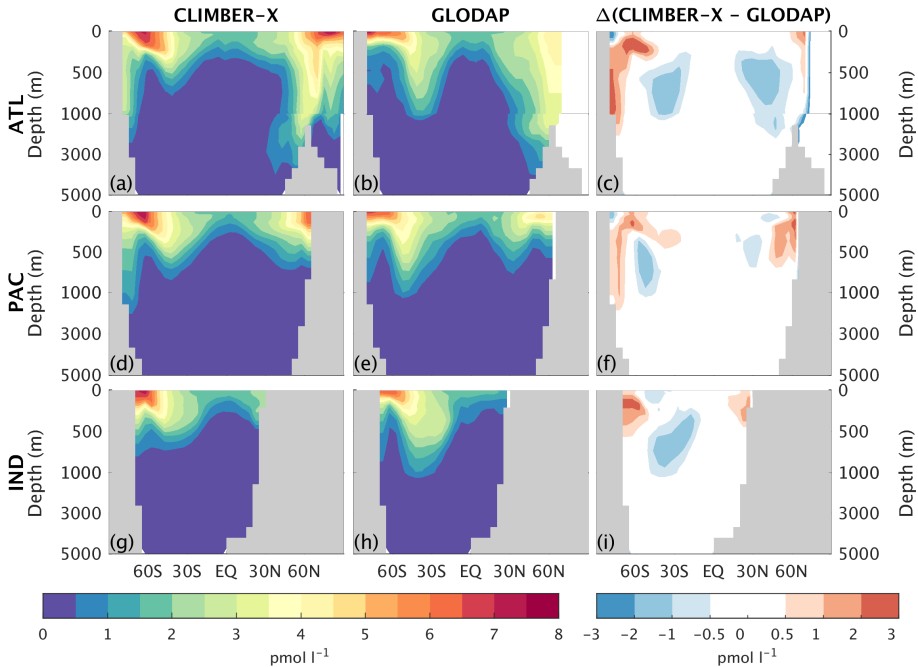

**Figure 5.** Zonally averaged CFC11 concentration for the year 1994 in CLIMBER-X (left column) and GLODAP (Key et al., 2004) (middle column) for different basins: Atlantic (top), Pacific (middle) and Southern Ocean (bottom). The model bias is shown in the right column.

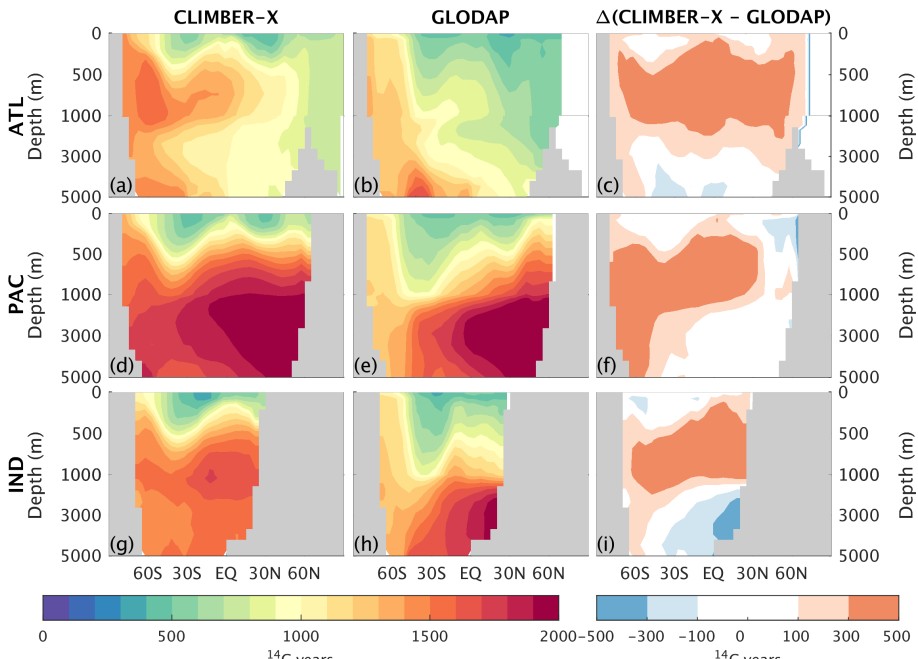

**Figure 6.** Zonally averaged pre-industrial radiocarbon ventilation age in CLIMBER-X (left column) and GLODAP Key et al. (2004) (middle column) for different basins: Atlantic (top), Pacific (middle) and Southern Ocean (bottom). The model bias is shown in the right column.

resolving synoptic processes in the atmosphere and therefore not representing the non-linear effects of synoptic variability on vertical mixing of tracers. For instance, a one-time mixing down to $200\,\mathrm{m}$ depth by a wind storm could have a large effect

on some tracers, which cannot be resolved by using climatological mean winds. We would expect this non-linear effect to be much more important for radiocarbon than for nutrients. The analyses of CFC11 and radiocarbon provide important insights on the ocean ventilation in the model and will be useful when discussing model biases in the distribution of other biogeochemical tracers below.

The spatial pattern of the air-sea $CO_2$ exchange is well captured by the model (Fig. 7), with outgassing generally taking

place in the tropics and $CO_2$ being taken up in mid- to high northern latitudes and in mid-latitudes of the Southern Hemisphere. The main difference compared to other models is observed around the Equator, with a less pronounced peak in $CO_2$ release simulated by CLIMBER-X (Fig. 8), which is likely related to deficiencies in the simulated ocean circulation close to the equator, where the geostrophic approximation employed in CLIMBER-X reaches its limit of applicability. In the Southern Ocean, most CMIP6 models tend to overestimate the $CO_2$ uptake compared to observations (e.g. Gruber et al., 2009) (Fig. 8),

while CLIMBER-X is apparently more consistent with recent estimates, although with substantial differences in the spatial distribution of the $CO_2$ flux (Fig. 7). Notably, in the Southern Ocean the CLIMBER-X air-sea $CO_2$ exchange diverges from that simulated by the MPI-ESM1-2-LR model (Fig. 8), which employs the original HAMOCC6 ocean biogeochemistry model. This is possibly related to the lower simulated net primary production in the Southern Ocean in CLIMBER-X compared to

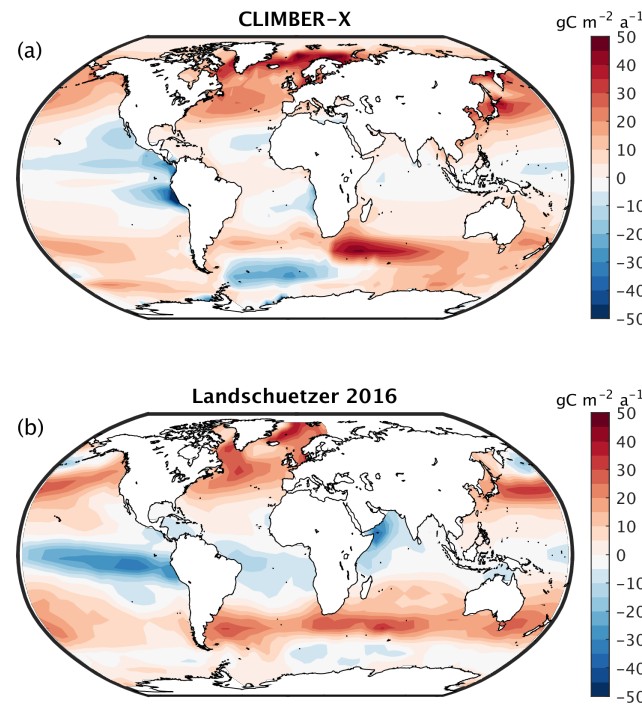

**Figure 7.** 1985-2010 average air–sea $CO_2$ flux in (a) CLIMBER-X compared to (b) observations from Landschützer et al. (2016).

MPI-ESM (Fig. 9a). However, the MPI-ESM seems to be an outlier in the simulated primary production in the Southern Ocean,
possibly because of biases in climate, which are unrelated to the HAMOCC ocean carbon cycle model.

The export of particulate organic carbon from the euphotic layer drives the biological pump and generally follows the primary productivity pattern, with modifications due to varying sinking speeds and remineralisation rates of POC in the water column. While the net primary productivity in CLIMBER-X is in line with CMIP6 models (Fig. 9a) and the globally integrated value of $55\,\mathrm{PgCyr^{-1}}$ agrees well with observations (Table 2), the export production in the model is generally at the lower end
of the CMIP6 model range (Fig. 9b). $CaCO_3$ and opal export are compared to CMIP6 models in Fig. 9c,d.

Primary production in the ocean is limited by the availability of nutrients. Over large parts of the surface ocean nitrogen concentrations constitute the main limiting factor for photosynthesis in CLIMBER-X (Fig. 10). However, over the Southern Ocean, in the equatorial Pacific and in the North Pacific production is limited by the availability of iron (Fig. 10). This is in accordance with observations showing that iron limitation is usually important where subsurface nutrient supply is enhanced,
such as in oceanic upwelling regions (e.g. Moore et al., 2013). Since one of the main iron sources in the ocean is from mineral dust deposited at the ocean surface (e.g. Tagliabue et al., 2016), iron limitation is confined to regions with low dust deposition. The dust cycle is an integral part of CLIMBER-X, and the dust deposition is therefore explicitly modelled. The simulated dust deposition compares reasonably well with estimates from complex ESMs for the present–day (Fig. 11), although they

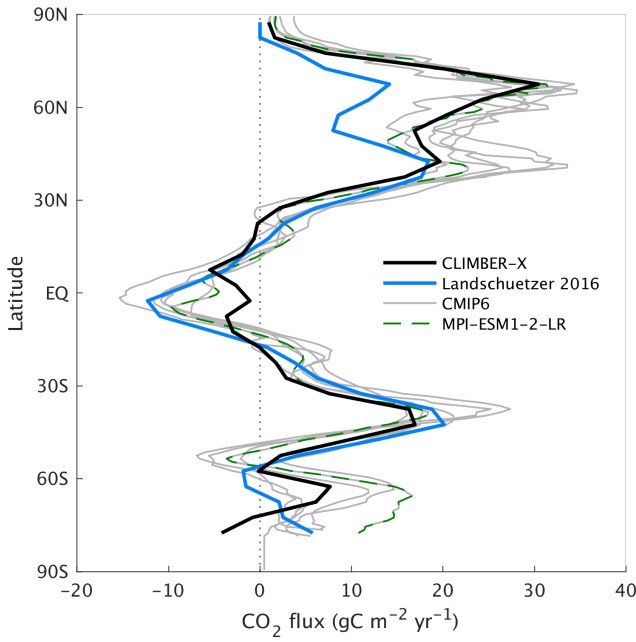

**Figure 8.** Zonal mean air–sea $CO_2$ flux (1985-2010 average) in CLIMBER-X compared to observations from Landschützer et al. (2016) and selected CMIP6 models.

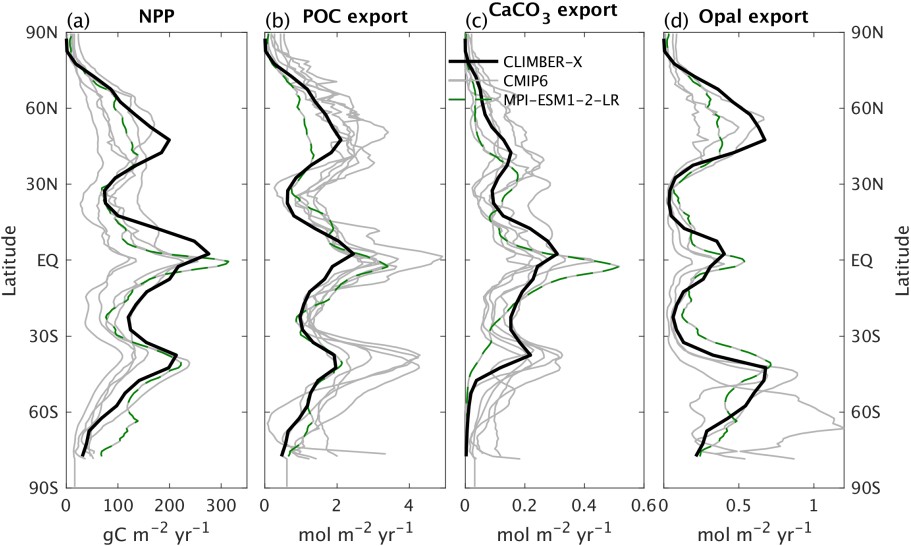

**Figure 9.** 1981-2010 average global zonal mean (a) net primary production, (b) particulate organic carbon export at 100 m depth, (c) $CaCO_3$ and (d) opal export at 100 m depth. Results from CLIMBER-X are compared to CMIP6 models.

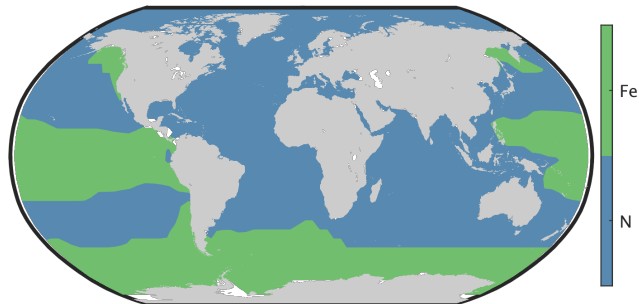

**Figure 10.** Nutrient limitation of marine net primary productivity in CLIMBER-X.

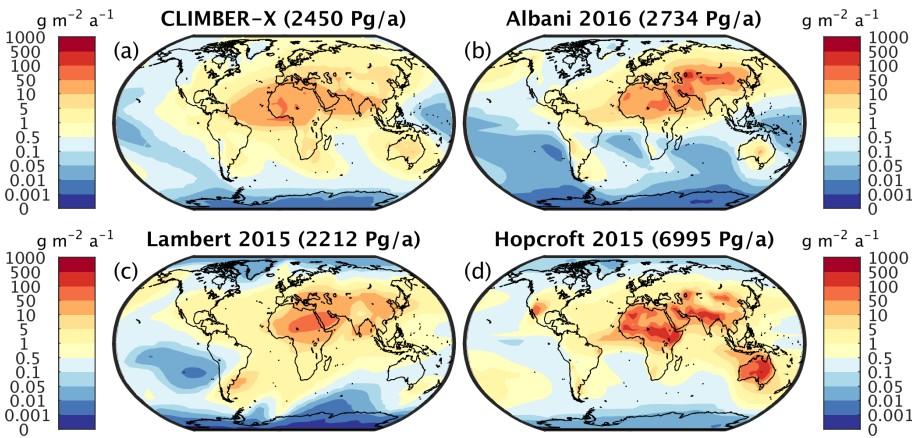

**Figure 11.** (a) CLIMBER-X annual dust deposition flux compared to model based products of (b) Albani et al. (2016), (c) Lambert et al. (2015) and (d) Hopcroft et al. (2015). The respective globally integrated deposition values are given in brackets in the panel titles.

are relatively poorly constrained. A comparison of dust deposition fluxes with observations over land further indicates that the
model is able to capture the general pattern of dust deposition rate (Fig. 12).

The simulated dissolved iron concentration in surface water is closely related to the dust deposition shown in Fig. 11. It is therefore high in the Atlantic and Indian oceans, lower in the Southern Ocean and very small over large parts of the Pacific (Fig. 13). This is broadly consistent with observations (e.g. Tagliabue et al., 2012), but measurements of iron concentration in ocean water are still relatively sparse.

The main features of the surface nitrate concentration are well reproduced by CLIMBER-X, with large concentrations in the Southern Ocean, moderate values in the upwelling region of the Eastern equatorial Pacific and in the North Atlantic and North Pacific and low values elsewhere (Fig. 15, Fig. 14a). The most pronounced model biases are found in too high nitrate concentrations in the Arctic and too low values in the North Pacific. The simulated basin-wide vertical distribution of nitrate is in very good agreement with observations (Fig. 16).

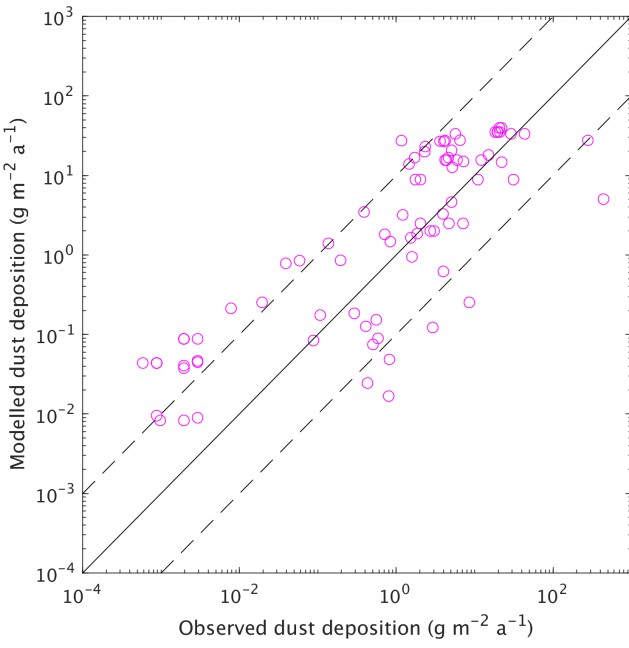

**Figure 12.** Simulated versus observed dust deposition fluxes at different locations available from the AeroCom dataset (Huneeus et al. (2011) and references therein). The dashed lines indicate one order of magnitude deviation from the 1:1 line.

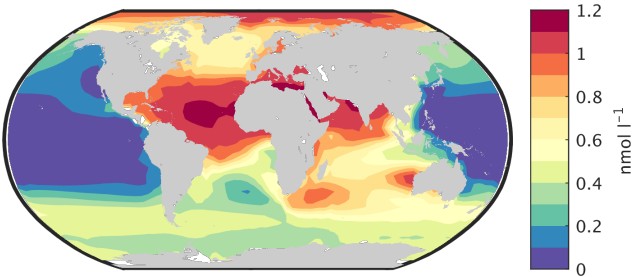

**Figure 13.** Average surface dissolved iron concentration in CLIMBER-X over the period 1981-2010.

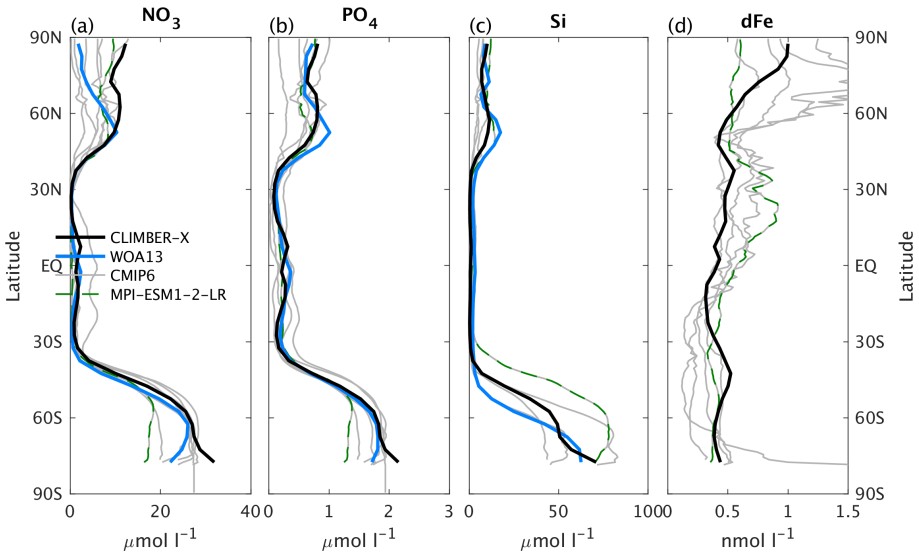

**Figure 14.** 1981-2010 average global zonal mean surface concentrations of the nutrients (a) nitrate, (b) phosphate, (c) silicate and (d) dissolved iron. CLIMBER-X is compared to observations (Garcia et al., 2013b) and CMIP6 model results.

The 3D phosphate distribution in the global ocean is nicely captured by the model (Fig. 17, Fig. 14b, Fig. 16), except for too low concentrations simulated in the surface ocean of the North Pacific and the northern Indian Ocean. The negative bias in the North Pacific is consistent with the too low simulated surface nitrate concentrations, both originating from a too vigorous ventilation of water masses in the upper kilometer in the physical ocean model.

As a result of reduced primary productivity in the Southern Ocean in CLIMBER-X compared to MPI-ESM1-2-LR, both sur-
face nitrate and phosphate concentrations are consistently higher in CLIMBER-X (Fig. 14a,b), as less nutrients are assimilated during photosynthesis.

Silicate concentration is generally overestimated in the sub-surface ocean and underestimated in the deep North Pacific and North Indian oceans (Fig. 18), similarly to other nutrients (Fig. 17).

The large scale patterns of oxygen concentration in ocean waters simulated by CLIMBER-X is largely consistent with
observations (Fig. 19), but the extent and depth of the oxygen minimum zones, in particular in the Eastern equatorial Pacific, is overestimated. This bias is common to many CMIP5 models (e.g. Cabré et al., 2015). Other biases include a too oxygen depleted Southern Ocean and too high oxygen concentrations in the upper North Pacific and North Indian oceans, again resulting from the excessive water mass ventilation in those regions as discussed above.

Both DIC and alkalinity are generally overestimated in the upper ocean (Fig. 16), particularly in Antarctic intermediate
water masses, and underestimated in the deep ocean (Fig. 20,21). These biases in the simulated vertical distribution of DIC and alkalinity could be due to a relatively low $CaCO3$ export from the euphotic layer (Table 2), which leads to a too weak vertical redistribution. Additionally, the simulated DIC concentration is generally too low in the North Pacific and northern Indian Oceans.

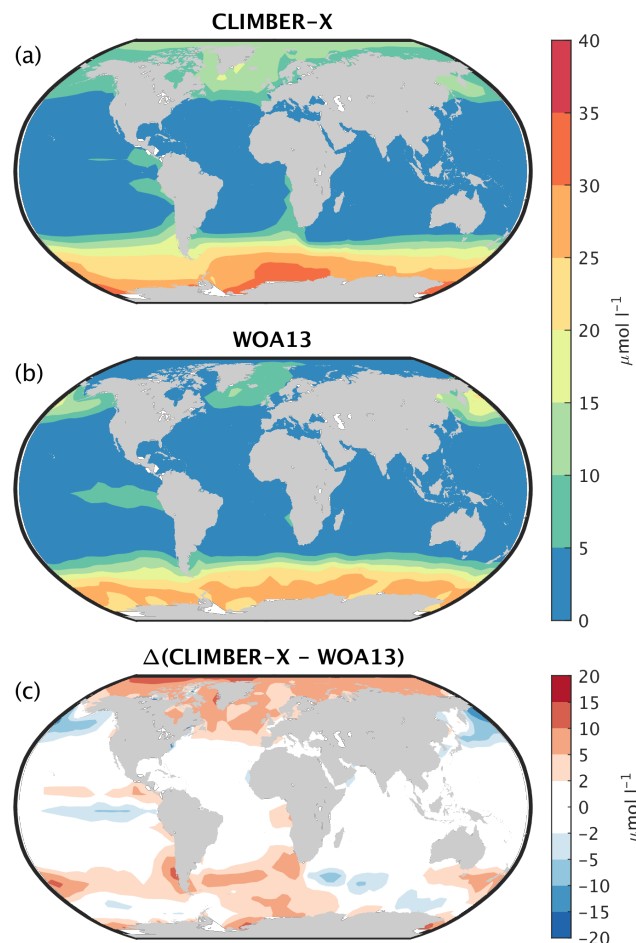

**Figure 15.** Surface $NO_3$ concentration in (a) CLIMBER-X (1981-2010 average) compared to (b) observations from the World Ocean Atlas 2013 (WOA13, Garcia et al. (2013b)). The model bias is shown in (c).

The carbon 13 isotope in the ocean helps to track the distribution of different water masses. The higher $\delta^{13}C$ values in the Atlantic compared to the Pacific Ocean, originating mainly from the pronounced overturning circulation in the Atlantic, which is absent in the Pacific, are generally captured by the model (Fig. 22). The negative biases at 500-1500 m depth are associated with the 'nutrient trapping' problem (Aumont et al., 1999; Dietze and Loeptien, 2013) that is often seen in ESMs. This problem is characterised by high concentrations of remineralised nutrients and carbon and, therefore, low $\delta^{13}C$ (Liu et al., 2021). The positive biases through the whole water column in the North Atlantic, North Pacific and northern Indian Ocean are possibly the result from too strong ventilation in these regions in the model.

In the Atlantic and Indian Ocean $CaCO_3$ dominates the sediment composition, in accordance with observations (Fig. 23a,d). However, little $CaCO_3$ is simulated in large parts of the sediment in the eastern Pacific Ocean, where observations indicate widespread $CaCO_3$ content in the Southern Hemisphere (Fig. 23a,d). The underestimation of calcite weight fractions in sedi-

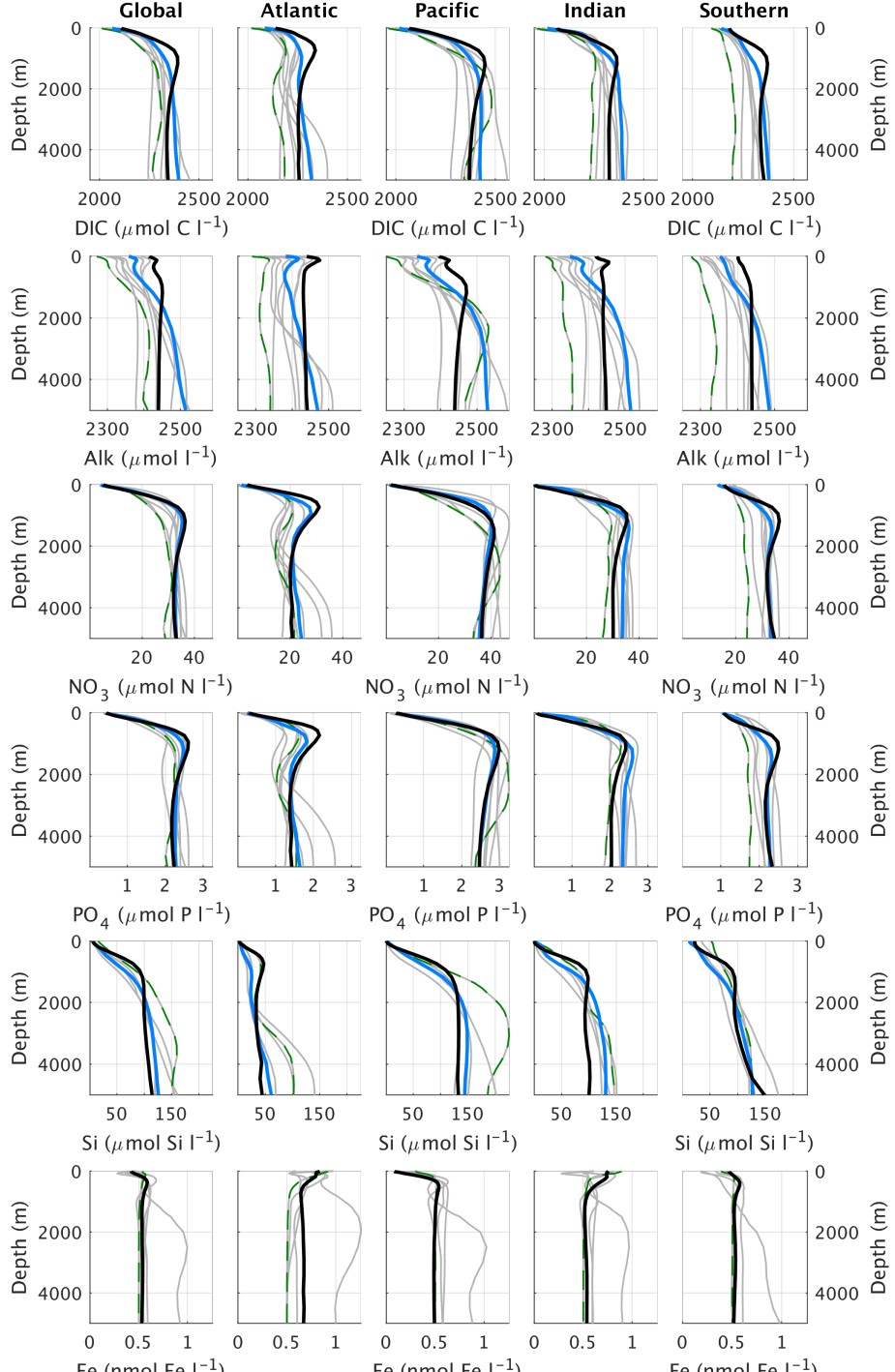

**Figure 16.** Global and basin-wide average profiles of different biogeochemical tracers in the ocean, from top to bottom: DIC, alkalinity, nitrate, phosphate, silicate and dissolved iron. CLIMBER-X results (black) are compared to observations (blue) (Lauvset et al., 2016; Olsen et al., 2016; Garcia et al., 2013a,b) and CMIP6 model results (grey). Results from the MPI-ESM1-2-LR are shown by the green dashed lines. The boundary of the Southern Ocean is set at 35 °S and the Southern Ocean section is not included in the profiles of the Atlantic, Pacific and Indian ocean. CLIMBER-X and CMIP6 data are averages over the time period 1981-2010.

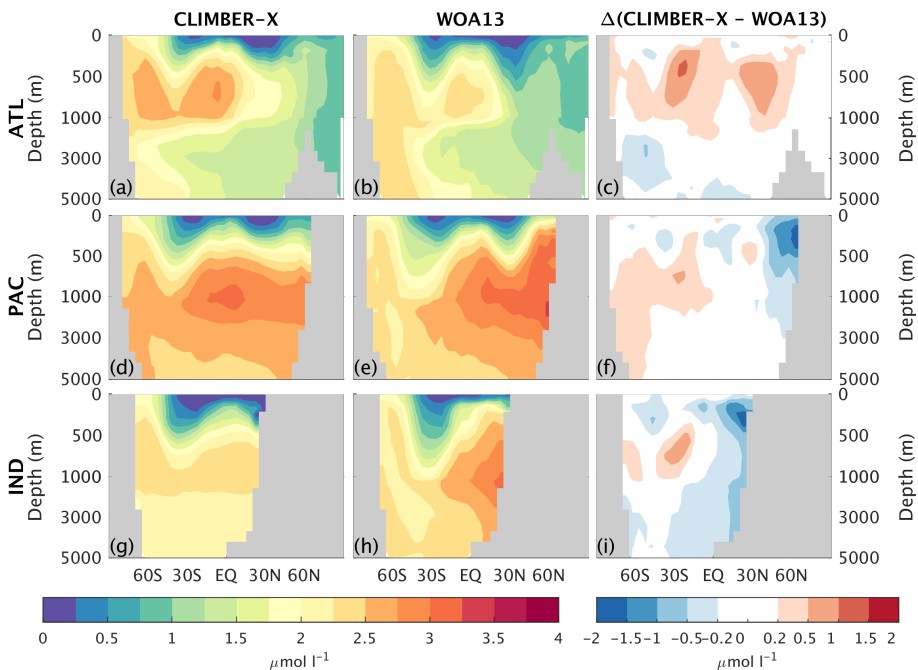

**Figure 17.** Zonally averaged $PO_4$ concentration in CLIMBER-X, 1981-2010 average, (left column) and WOA13 (Garcia et al., 2013b) (middle column) for different basins: Atlantic (top), Pacific (middle) and Southern Ocean (bottom). The model bias is shown in the right column.

ments of the eastern South Pacific Ocean is caused by water being undersaturated with respect to calcite in this area. This leads
to dissolution of most of the calcite produced at the surface before it can even reach the sediments. The strongly undersaturated
water is ultimately a result of deficiencies in the simulated ocean circulation. Some other models show similar deficiencies in
the simulated calcite fraction in Pacific sediments (e.g. Kurahashi-Nakamura et al., 2022). Global $CaCO_3$ sediment deposi-
tion and burial are in line with observational underestimates (Table 2), with around 25 % of the deposited $CaCO_3$ undergoing
dissolution. The opal content in sediments in CLIMBER-X is overestimated (Fig. 23b,e), even though the global opal sedimen-
430 tation and burial fluxes are fully consistent with observational estimates (Table 2). Opal is particularly abundant in the eastern
equatorial Pacific, simply as a result of missing $CaCO_3$ in the sediments in that area. Organic carbon is found mainly on the
continental margins and in the equatorial east Pacific, in agreement with observations (Fig. 23c,f), although CLIMBER-X tends
to underestimate the organic carbon content in sediments, possibly because of a too small sediment deposition flux of POC
(Table 2).

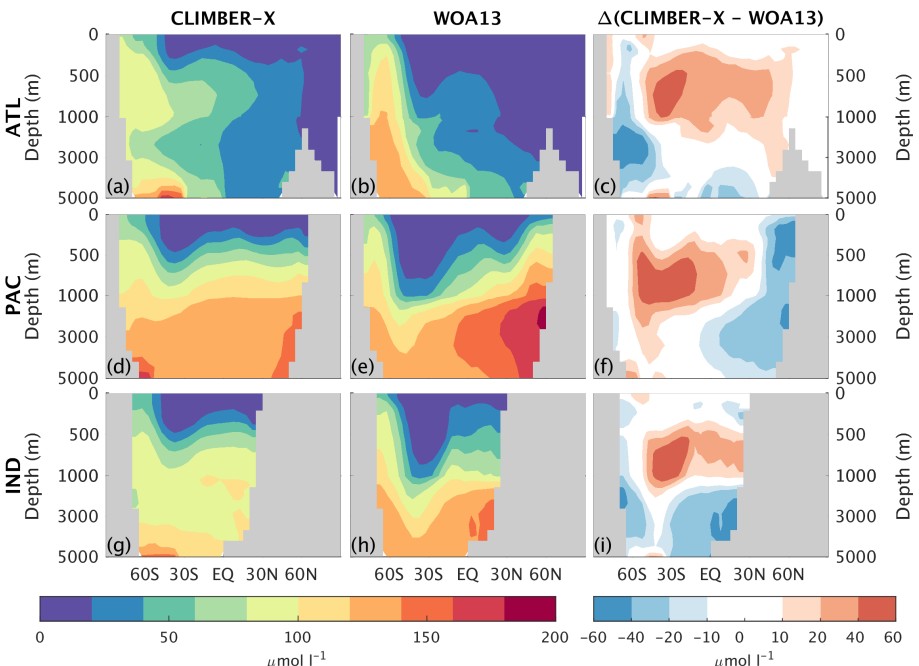

**Figure 18.** Zonally averaged Si concentration in CLIMBER-X, 1981-2010 average, (left column) and WOA13 (Garcia et al., 2013b) (middle column) for different basins: Atlantic (top), Pacific (middle) and Southern Ocean (bottom). The model bias is shown in the right column.

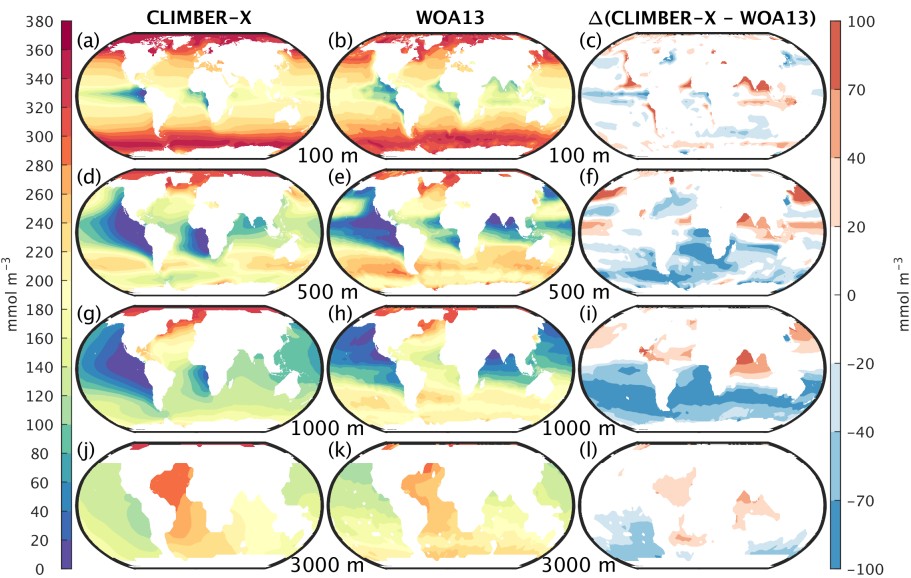

**Figure 19.** Oxygen concentration in CLIMBER-X, 1981-2010 average, (left column) and WOA13 (Garcia et al., 2013a) (middle column) at different ocean depths: from top to bottom, 100 m, 500 m, 1000 m and 3000 m. The model bias is shown in the right column.

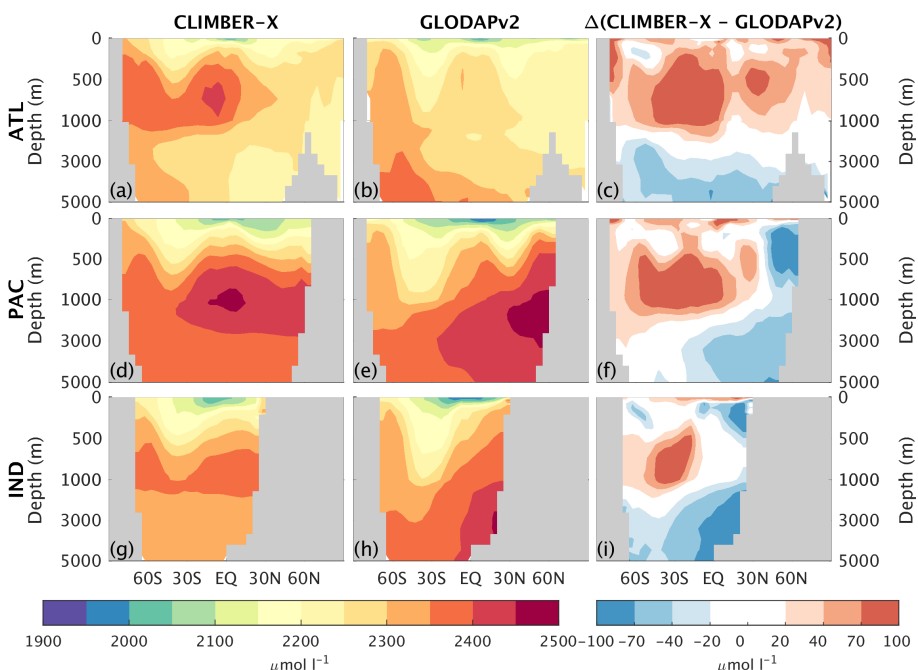

**Figure 20.** Zonally averaged dissolved inorganic carbon in CLIMBER-X, 1981-2010 average, (left column) and GLODAPv2 (Lauvset et al., 2016; Olsen et al., 2016) (middle column) for different basins: Atlantic (top), Pacific (middle) and Southern Ocean (bottom). The model bias is shown in the right column.

### 4.1.2 Land carbon cycle

A detailed evaluation of the land carbon cycle component has already been presented in the original PALADYN description paper (Willeit and Ganopolski, 2016). However, here we partly repeat the analysis to show the model performance in the coupled climate model setup and with the additional modifications to the model described above.

A selection of simulated global variables characterising the land carbon cycle is presented and compared to observation-based estimates in Table 3, providing a summary of model performance for the present day.

Photosynthesis is the basic process by which carbon enters the land domain. The simulated gross primary production (GPP), which quantifies this process, is in good agreement with observational estimates, both in terms of global integral (Table 3) and in terms of spatial distribution (Fig. 24a,b,c).

The total carbon stored in the vegetation, both above ground and below ground, is slightly overestimated in the model (Table 3), but the meridional distribution, mainly originating from large-scale differences in precipitation, is well reproduced (Fig. 24d,e,f). Most of soil carbon in CLIMBER-X is stored in cold soils of the NH high-latitudes, in agreement with observations (Fig. 24g,h,i). However, compared to estimates from Carvalhais et al. (2014) the soil carbon distribution is too skewed towards high northern latitudes and there is too little carbon in the tropics. Most CMIP6 models underestimate soil carbon in the tropics as well (Fig. 24j).

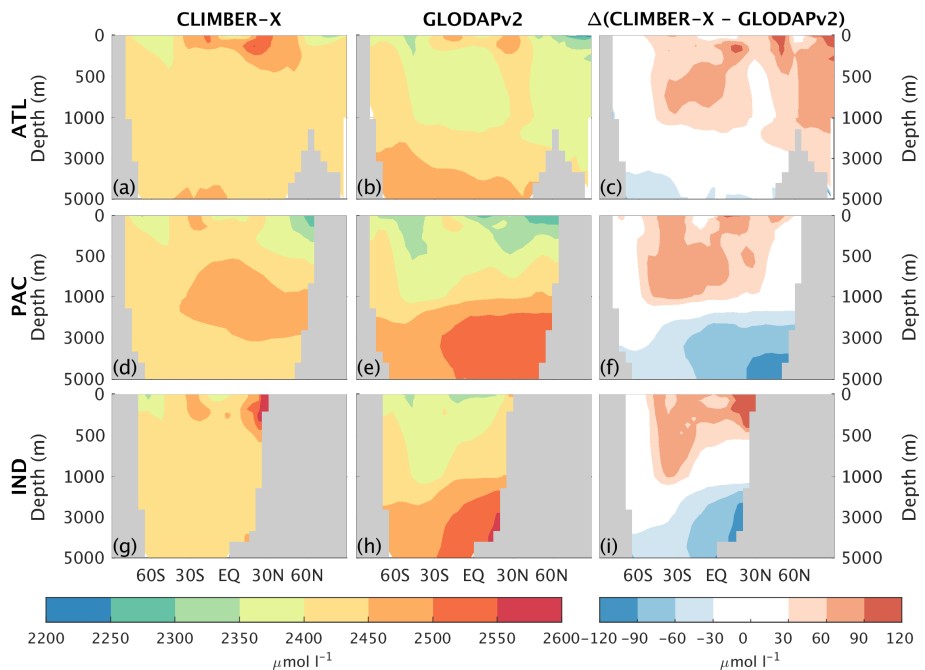

**Figure 21.** Zonally averaged total alkalinity in CLIMBER-X, 1981-2010 average, (left column) and GLODAPv2 (Lauvset et al., 2016; Olsen et al., 2016) (middle column) for different basins: Atlantic (top), Pacific (middle) and Southern Ocean (bottom). The model bias is shown in the right column.

In CLIMBER-X, $\sim$1500 PgC of carbon are stored in the top soil meter in good agreement with different estimates (Table 3). However, with $\sim$2150 PgC, the total soil carbon content seems to be underestimated compared to observations, which suggest >3000 PgC. This indicates that too little carbon is simulated in soil below 1 m depth. However, total soil carbon content estimates vary widely between datasets (e.g. Fan et al., 2020), with e.g. 1952$\pm$198 PgC in WISE30sec (Batjes, 2016) and 3141$\pm$893 PgC in Sanderman et al. (2017) in the top 2 m of soil. Most of carbon in mineral soil layers below one meter is recalcitrant and its response to changes in environmental conditions is uncertain. In Earth System models, total soil carbon storage is usually much lower (1206$\pm$445 PgC, Varney et al. (2022)) as these models account for active carbon responding on centennial time scale. In CLIMBER-X, one possible explanation for underestimated carbon content in deeper soil layers is that the maximum turnover time scale of soil carbon is set to 5000 years in the model, which limits the amount of carbon that can be accumulated in cold, frozen soil layers. Other possible reasons include: (i) a general underestimation of vertical carbon transport by diffusion, particularly into perennially frozen soil layers, (ii) a possible depth dependence of soil carbon turnover due to processes other than temperature and moisture (e.g. Koven et al., 2013) and that are not included in the model. Consistently, the carbon contained in areas affected by permafrost is $\sim$800 PgC, which is also a bit lower than the $\sim$1100–1500 PgC suggested by observations (Table 3). Let us note that even when models are initialized with the observed permafrost carbon stock of $\approx$1300 PgC, remapping on model resolution and accounting for differences in soil temperatures

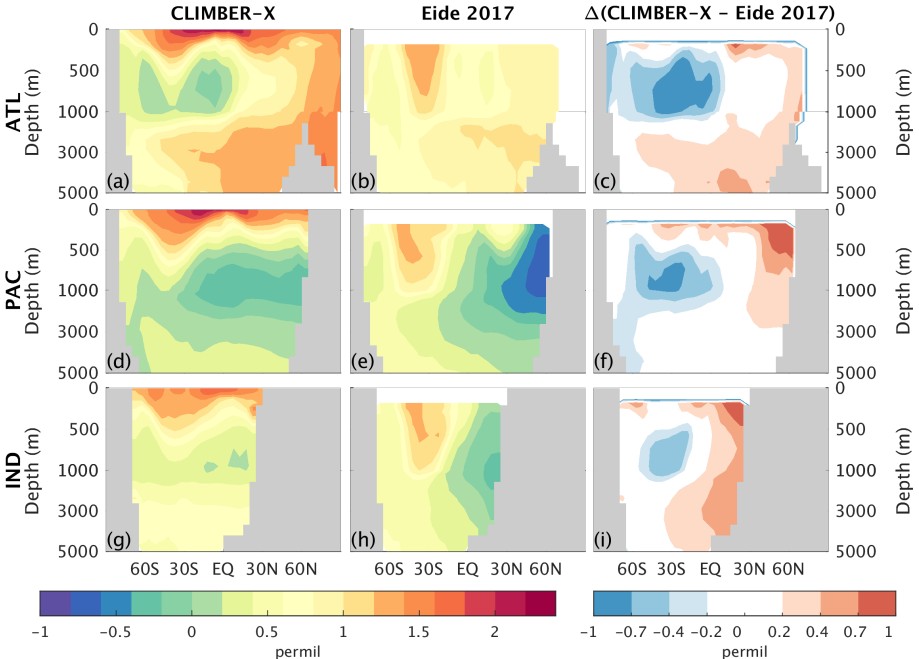

**Figure 22.** Zonally averaged $\delta^{13}$C in CLIMBER-X, 1981-2010 average, (left column) and Eide et al. (2017) (middle column) for different basins: Atlantic (top), Pacific (middle) and Southern Ocean (bottom). The model bias is shown in the right column.

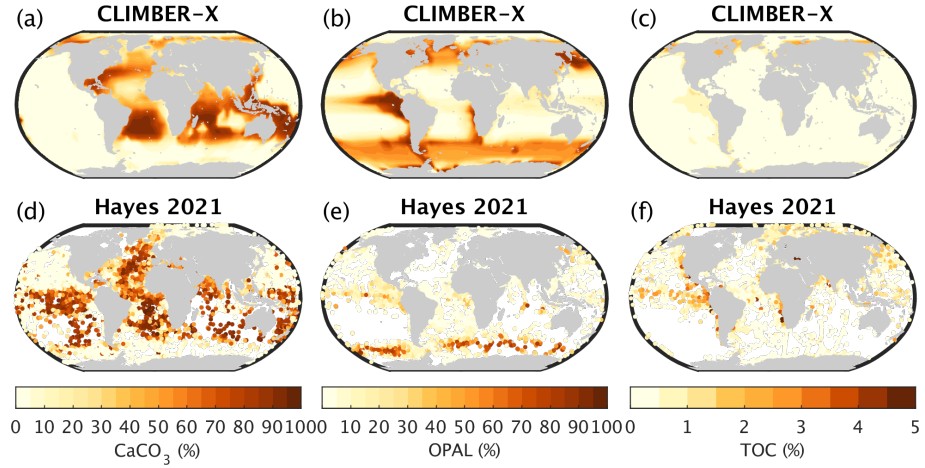

**Figure 23.** Weight fraction of calcite, opal and organic carbon in marine sediments as simulated by CLIMBER-X (top), compared to observations (Hayes et al., 2021) (bottom).

**Table 3.** Global values for the main variables of the land carbon cycle.

| | CLIMBER-X | Estimated range | Unit | Source |
|---|---|---|---|---|
| *Primary production* | | | | |
| Gross primary production | 120 | 115–131 | $PgCyr^{-1}$ | Beer et al. (2010) |
| Net primary production | 67 | 42–70 | $PgCyr^{-1}$ | Ito (2011) |
| *Land carbon pools* | | | | |
| Vegetation carbon | 472 | 392–437 | PgC | Fan et al. (2020) |
| Soil carbon | 2145 | 3300–4800 | PgC | Fan et al. (2020) |
| Soil carbon top 1 m | 1521 | 1200–2000 | PgC | Varney et al. (2022) |
| Soil carbon top 1 m 60–90°N | 436 | 314–526 | PgC | Varney et al. (2022) |
| Permafrost area | 19.1 | 18.7 | $10^6 \times km^2$ | Brown et al. (1998);Tarnocai et al. (2009) |
| Carbon in permafrost area | 796 | 1100–1500 | PgC | Hugelius et al. (2014) |
| Peatland area | 2.4 | 4.4 | $10^6 \times km^2$ | Yu et al. (2010) |
| Carbon in peatlands | 340 | 530–694 | PgC | Yu et al. (2010) |
| $CH_4$ | | | | |
| Maximum monthly wetland area | 5 | 5.1 | $10^6 \times km^2$ | Prigent et al. (2007);Papa et al. (2010) |
| Total $CH_4$ emissions | 214 | 100–217 | $TgCH_4yr^{-1}$ | Saunois et al. (2020) |
| Tropical $CH_4$ emissions | 182 | 71–155 | $TgCH_4yr^{-1}$ | Saunois et al. (2020) |
| Extratropical $CH_4$ emissions | 33 | 12–64 | $TgCH_4yr^{-1}$ | Saunois et al. (2020) |
| *Weathering (pre-industrial)* | | | | |
| $CO_2$ consumption | 22.6 | 17–27 | $TmolCyr^{-1}$ | Munhoven (2002) |
| Carbonate weathering | 20.1 | 10–25.4 | $TmolCyr^{-1}$ | Munhoven (2002) |
| Silicate weathering | 12.6 | 10.8–19.7 | $TmolCyr^{-1}$ | Munhoven (2002) |
| Alkalinity flux to ocean | 32.7 | 30–40 | $Tmolyr^{-1}$ | Amiotte Suchet et al. (2003); Gaillardet et al. (1999) |

between models and observations generally leads to a reduction of permafrost carbon stocks (e.g. Kleinen and Brovkin, 2018). The CLIMBER-X simulated peatland extent is lower than estimated (Yu et al., 2010), and consistently also the peat carbon is underestimated accordingly (Table 3).

    The turnover time of terrestrial ecosystem carbon is an integrated quantitative measure of the residence time of carbon on land, from the time it is fixed by photosynthesis to the time it is returned to the atmosphere through respiration processes. It 470   is computed as the ratio between land carbon stocks (vegetation+soil) and gross primary production. The ecosystem carbon turnover time simulated by CLIMBER-X is in line with CMIP6 models, while it is underestimated compared to observation-based estimates from Fan et al. (2020) (Fig. 24j,k,l). However, it should be noted that the large uncertainties in soil carbon content result in a rather uncertain estimated ecosystem carbon turnover time (Fan et al., 2020).

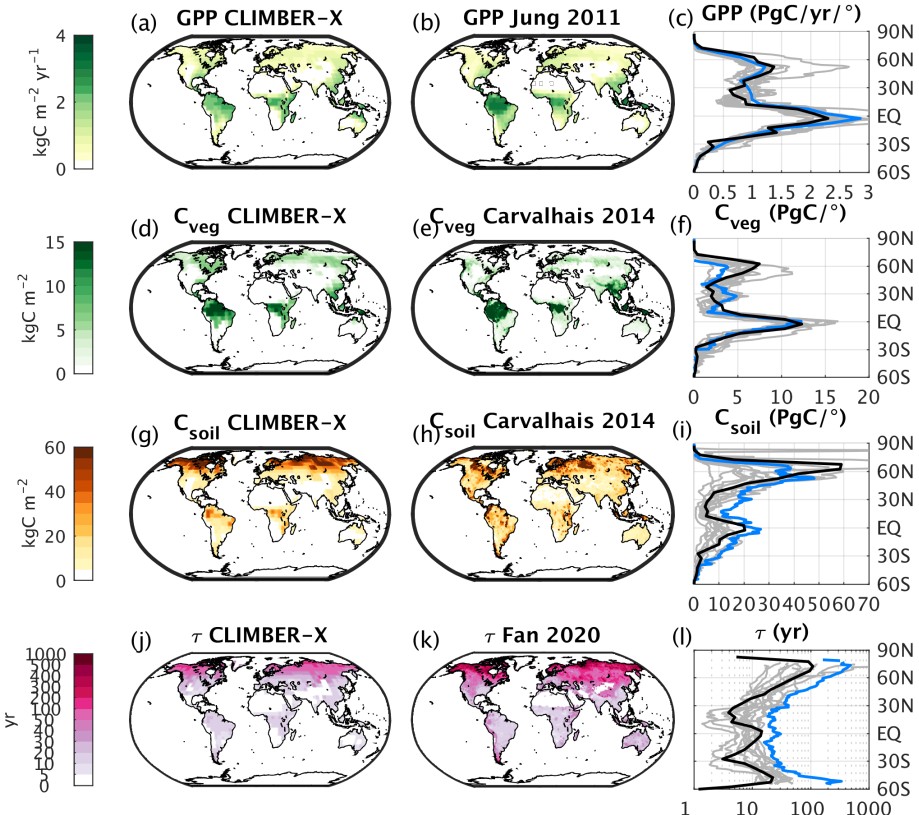

**Figure 24.** (a) Simulated GPP compared to (b) observations (Jung et al., 2011). (c) Comparison of zonally integrated GPP. (d) Simulated vegetation carbon compared to (e) observations (Carvalhais et al., 2014). (f) Comparison of zonally integrated vegetation carbon. (g) Simulated soil carbon compared to (h) observations (Carvalhais et al., 2014). (i) Comparison of zonally integrated soil carbon. (j) Simulated ecosystem carbon turnover time compared to (k) observations (Fan et al., 2020). (l) Comparison of zonal mean ecosystem carbon turnover time. In panels (c), (f), (i) and (l) results from CLIMBER-X are shown in black, observations in blue and CMIP6 models in grey. CLIMBER-X and CMIP6 data are averages over the time period 1981-2010.

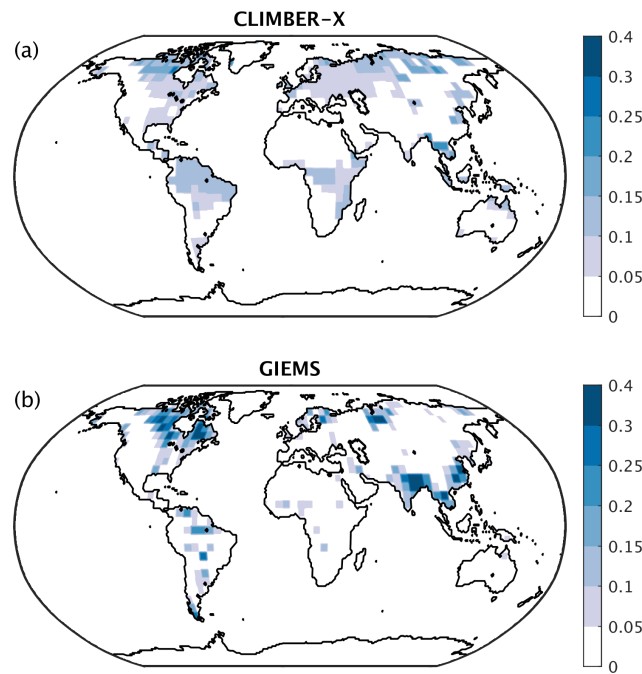

**Figure 25.** Maximum monthly wetland fraction (a) in CLIMBER-X compared to (b) the GIEMS dataset (Papa et al., 2010; Prigent et al., 2007).

The global maximum monthly wetland extent in CLIMBER-X agrees well with observations (Table 3), although with sub-
475 stantial differences in the geographic distribution (Fig. 25). Compared to the multi-satellite product from GIEMS (Global
Inundation Extent from Multi-Satellites) (Prigent et al., 2007; Papa et al., 2010) the model simulates larger wetland extent
in tropical forest areas. However, if compared to other wetland products based on data other than from satellite, GIEMS is
underestimating wetlands below dense forests (e.g. the Amazon forest) (e.g. Melack and Hess, 2010) In south-east Asia, the
GIEMS wetland extent also includes extensive rice cultivation areas, which are not represented in the model.
In CLIMBER-X methane is emitted exclusively from wetlands. However, because of the dependence of methane emissions
on soil carbon decomposition rates and because of the temperature dependence of the fraction of wetland carbon respired as
methane, wetland methane emissions are dominated by tropical sources (Table 3, Fig. 26), in agreement with observations (e.g.
Saunois et al., 2020). The total $CH_4$ emissions from wetlands are at the high end of recent estimates, which is a result of tuning
the emissions in the model to match the observed emissions from all natural sources.
Chemical weathering fluxes are generally high where runoff is high, with the separation between silicate and carbonate
weathering being modulated by lithological properties (Fig. 27). The global $CO_2$ consumption rate by weathering and the
alkalinity flux to the ocean in form of bicarbonate produced by rock weathering are in good agreement with observational
estimates (Table 3), while the partitioning between carbonate and silicate weathering is skewed toward carbonate weathering
(Table 3).

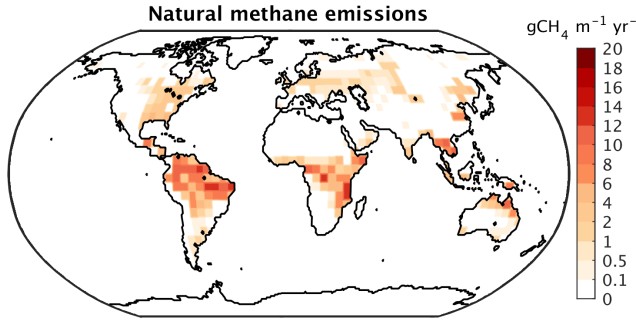

**Figure 26.** Natural methane emission simulated by CLIMBER-X for the present–day (1981-2010 average).

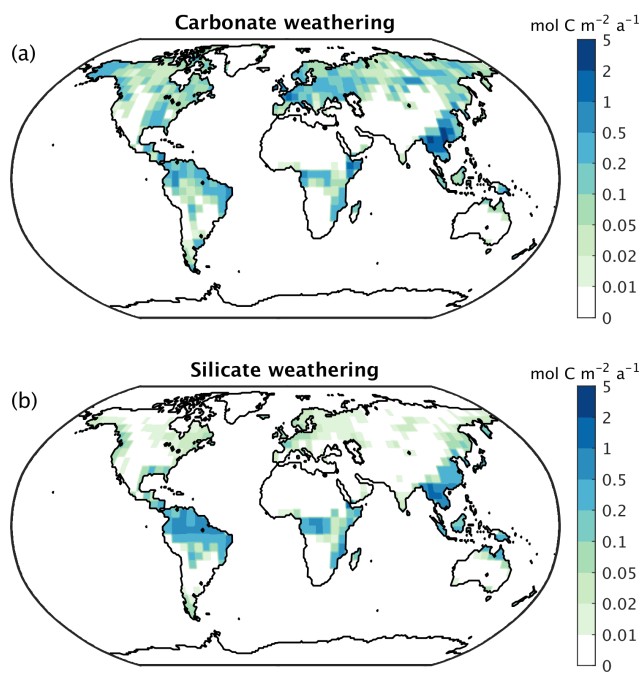

**Figure 27.** CLIMBER-X (a) silicate and (b) carbonate weathering flux distribution for the present–day (1981-2010 average).

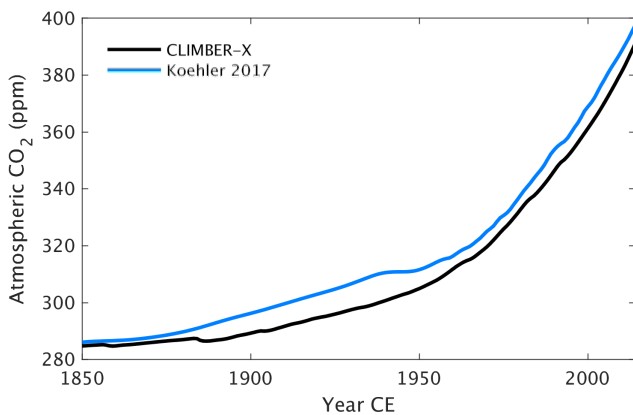

**Figure 28.** Historical atmospheric $CO_2$ concentration from a coupled CLIMBER-X simulation compared to observations (Köhler et al., 2017).

## 490  4.2  Historical period

As shown by Willeit et al. (2022), the historical climate evolution is well simulated by CLIMBER-X. Here we extend this analysis by focusing on the carbon cycle response.

The historical atmospheric $CO_2$ concentration is well reproduced by the model, with $CO_2$ at the year 2015 being within $\sim$5ppm of direct measurements (Fig. 28). Biases in simulated $CO_2$ of $\sim$10 ppm are quite common in state-of-the-art ESMs
(e.g. Hoffman et al., 2014; Friedlingstein et al., 2014).

The partitioning of the anthropogenic carbon emitted over the historical period among the different spheres is compared with recent estimates of the Global Carbon Budget (GCB) (Friedlingstein et al., 2022) by the Global Carbon Project in Fig. 29. The amount of fossil carbon emitted from anthropogenic activities is prescribed from empirical data and therefore by definition matches with estimates from Friedlingstein et al. (2022). The carbon emissions resulting from land use change practices are
underestimated in CLIMBER-X compared to the GCB, although the actual values remain uncertain (e.g. Gasser et al., 2020). A substantial fraction of this anthropogenic $CO_2$ emission is absorbed by the ocean and the land, while the rest remains in the atmosphere. In CLIMBER-X, the ocean carbon uptake is a bit lower and the land carbon uptake a bit higher than GCB estimates, but the net effect is a realistic airborne fraction of carbon remaining in the atmosphere. The ocean carbon uptake is driven by the chemical disequilibrium between surface air $CO_2$ concentrations and the concentration of dissolved $CO_2$ in
the surface ocean water and is relatively well understood, as also indicated by the narrow uncertainty range obtained from different CMIP6 models (Fig. 30a). The CLIMBER-X ocean carbon uptake falls within this narrow range although it tends to be at the lower end. The land carbon uptake is largely driven by an increase in gross primary productivity as a response to increasing atmospheric $CO_2$. The net primary productivity increase simulated by CLIMBER-X over the historical period is in agreement with what is shown by most CMIP6 models (Fig. 30b). However, the effect of this NPP increase on vegetation
carbon varies widely among models (Fig. 30c), also because of the confounding factor of land use change. In CLIMBER-X the

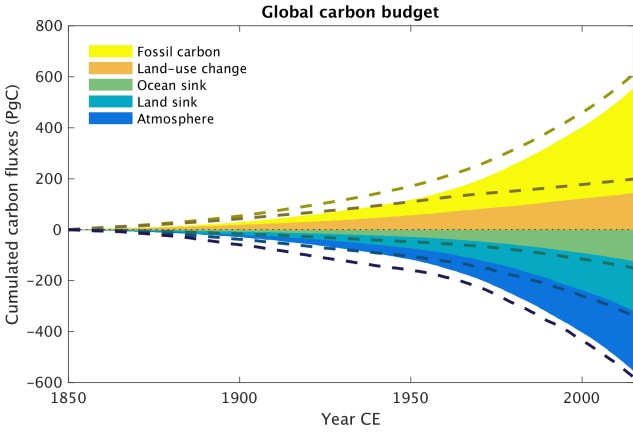

**Figure 29.** Historical global carbon budget in CLIMBER-X. The dashed lines are estimates from Friedlingstein et al. (2022).

net effect is a vegetation carbon stock decrease by $\sim$50 PgC. The historical evolution of soil carbon additionally depends on the response of microbial decomposition to changing environmental conditions, particularly soil temperatures. The increasing NPP, and consequently larger input of litter carbon into the soil, dominates over the negative effect of increasing temperatures in CLIMBER-X, leading to an increase in soil carbon by $\sim$50 PgC (Fig. 30d).

Since CLIMBER-X is enabled with carbon isotopes, it also allows for a comparison of isotopic signatures to observations, thereby providing additional constraints on processes involved in carbon cycle exchanges. As an example, the historical $\delta^{13}C$ of atmospheric $CO_2$ is compared to observations in Fig. 31.

The general historical trend in atmospheric $CH_4$ is captured by the model (Fig. 32a). Prescribed anthropogenic methane emissions are the dominant source for the increase of the atmospheric methane burden, but natural emissions from land are

also increasing due to the increase in NPP and soil temperature (Fig. 32b).

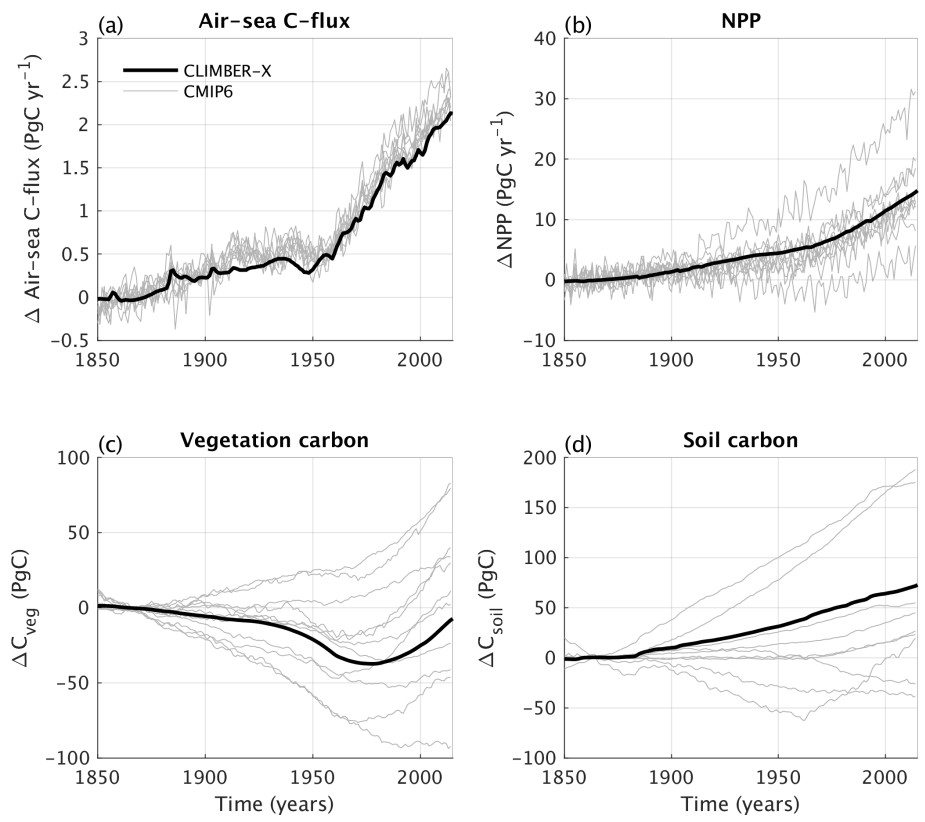

**Figure 30.** Historical anomalies of (a) air-sea $CO_2$ flux, (b) net primary production on land, (c) vegetation carbon and (d) soil carbon in CLIMBER-X compared to CMIP6 models. The anomalies are computed relative to the time interval 1850–1880 CE.

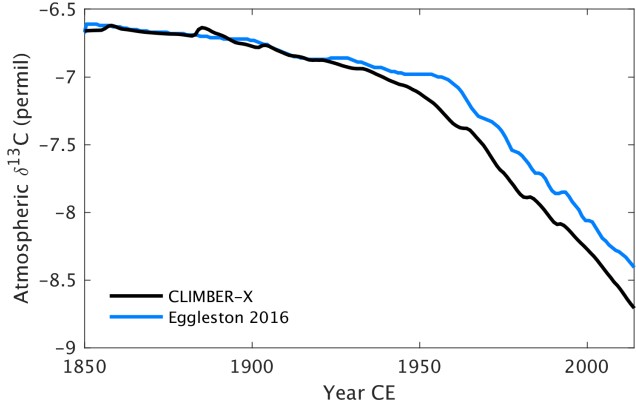

**Figure 31.** Historical $\delta^{13}C$ of atmospheric $CO_2$ in CLIMBER-X compared with observations (Eggleston et al., 2016).

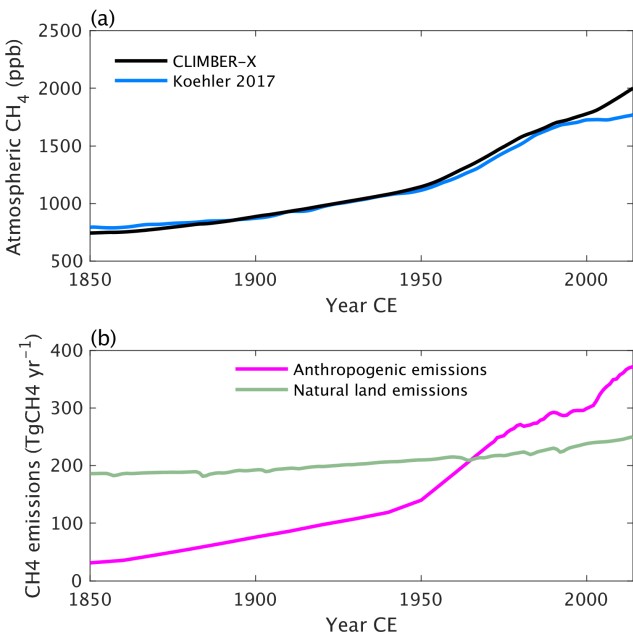

**Figure 32.** Historical (a) atmospheric $CH_4$ concentration in CLIMBER-X compared to observations (Köhler et al., 2017) and (b) prescribed anthropogenic $CH_4$ emissions and natural land emissions as simulated in CLIMBER-X.

## 5 Carbon cycle feedbacks

Carbon cycle processes both on land and in the ocean are sensitive to changes in climate and atmospheric $CO_2$. This implies that carbon fluxes between ocean and atmosphere and between land and atmosphere will respond to changes in climate and $CO_2$ concentration, which will in turn affect $CO_2$ and consequently climate. Quantifying the strength of these feedbacks is important to understand how the climate will respond to anthropogenic $CO_2$ emissions. For that, a linear feedback decomposition analysis has been proposed by Friedlingstein et al. (2006) to estimate these feedbacks in Earth system models. The analysis relies on a set of model simulations under idealized $1\,\%\mathrm{yr}^{-1}$ $CO_2$ increase experiments, whereby in one simulation the $CO_2$ increase is seen only by the radiative code in the atmosphere (radiatively coupled), in another one the $CO_2$ increase is seen only by the carbon cycle (biogeochemically coupled) and in a final one both radiation and carbon cycle see the increase in atmospheric $CO_2$ (fully coupled). This set of simulations allows to roughly separate the effect of changes in climate and changes in $CO_2$ on land and ocean carbon fluxes. To a first approximation, the carbon cycle feedback to climate is usually quantified using global temperature changes, while in reality climate obviously influences the carbon cycle in more complex ways than just through (global) temperature.

The carbon cycle feedback parameters have been estimated for C4MIP, CMIP5 and CMIP6 models (Friedlingstein et al., 2006; Arora et al., 2013, 2020). In Fig. 33 the carbon cycle–climate ($\gamma$) and the carbon cycle–concentration ($\beta$) feedbacks in CLIMBER-X are compared to CMIP6 model results separately for land and ocean. An increase in $CO_2$ has a positive effect on the uptake of carbon by both land and ocean, resulting in a lowering of $CO_2$ and therefore a negative feedback on climate (implying positive $\beta$, Fig.33a,c). Conversely, climate warming has a generally negative impact on the ability of ocean and land to store carbon, leading to a positive feedback loop (implying negative $\gamma$, Fig.33b,d). The $\beta$ and $\gamma$ values obtained here are well within the range of estimates from CMIP6 models (Arora et al., 2020), although in CLIMBER-X the $CO_2$ fertilisation effect on land is rather high (Fig.33a) and the ocean carbon uptake as a response to an increase in atmospheric $CO_2$ is at the lower end of the CMIP6 models (Fig.33c).

## 6 The zero emissions commitment

The zero emissions commitment (ZEC) is a measure of the amount of additional future temperature change following a complete cessation of $CO_2$ emissions (e.g. Matthews and Solomon, 2013). A model intercomparison project has been established in an effort to analyze and compare the ZEC in different Earth system models (Jones et al., 2019). Here we use this standardized and idealized experimental setup to compare the carbon cycle response in CLIMBER-X with results from the ZECMIP models for the 1000 PgC emission pulse (MacDougall et al., 2020). The experiment branches off from a $1\,\%$ per year $CO_2$ increase run with $CO_2$ emissions set to zero at the point of 1000 PgC of total carbon emissions. We performed this experiment with both the open and closed carbon cycle setups.

The results of the CLIMBER-X simulations are generally well within the large range of results from state-of-the-art ESMs and EMICs participating in ZECMIP (Fig. 34). Atmospheric $CO_2$ concentration is rapidly decreasing after stopping the carbon emissions (Fig. 34b), while global temperature shows a more modest decrease (Fig. 34a). The ocean continues to take up carbon

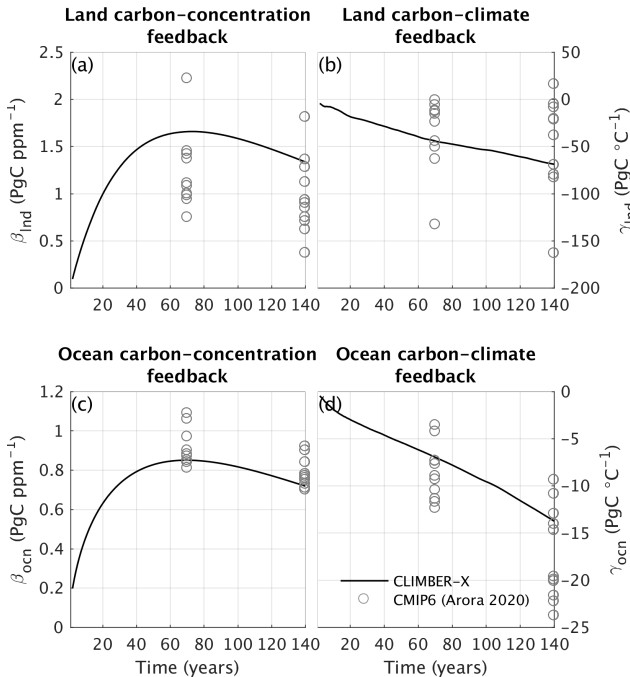

**Figure 33.** Carbon cycle feedbacks in CLIMBER-X compared to CMIP6 models (Arora et al., 2020). The (a,c) $\beta$ and (b,d) $\gamma$ parameters are shown at the time of $CO_2$ doubling (year $\sim$70) and at the time of $CO_2$ quadrupling (year $\sim$140) in idealized $1\,\%\mathrm{yr}^{-1}$ $CO_2$ increase experiments.

throughout the simulation (Fig. 34c), while the land turns from a sink to a source of carbon roughly at the time of the peak in 555 $CO_2$ (Fig. 34d). CLIMBER-X initially shows a relatively weak ocean carbon uptake compared to ZECMIP models, while the land carbon uptake is at the high end of the ZECMIP model ensemble.

The difference between the experiments with open and closed carbon cycle setup are negligible for the first few centuries but continue to grow over time, with $CO_2$ decreasing faster in the open setup (Fig. 34b).

## 7 Conclusions

We have described the major features of the carbon-cycle components of the newly developed CLIMBER-X Earth System model. The model includes a detailed representation of carbon cycle processes on land, in the ocean and in marine sediments, thus allowing to investigate the complex interactions between climate and the carbon cycle. Two setups of the global carbon cycle, closed and open, are available in CLIMBER-X, allowing both a proper comparison with CMIP6 type models on centennial scale and multi-millennia simulations. We have evaluated the model performance for the historical period against observations 565 and state-of-the-art CMIP6 models, showing that many characteristics and feedbacks of the carbon cycle are reasonably well captured by the model. Biases in the simulated distribution of ocean biogeochemical tracers exist and can mostly be related

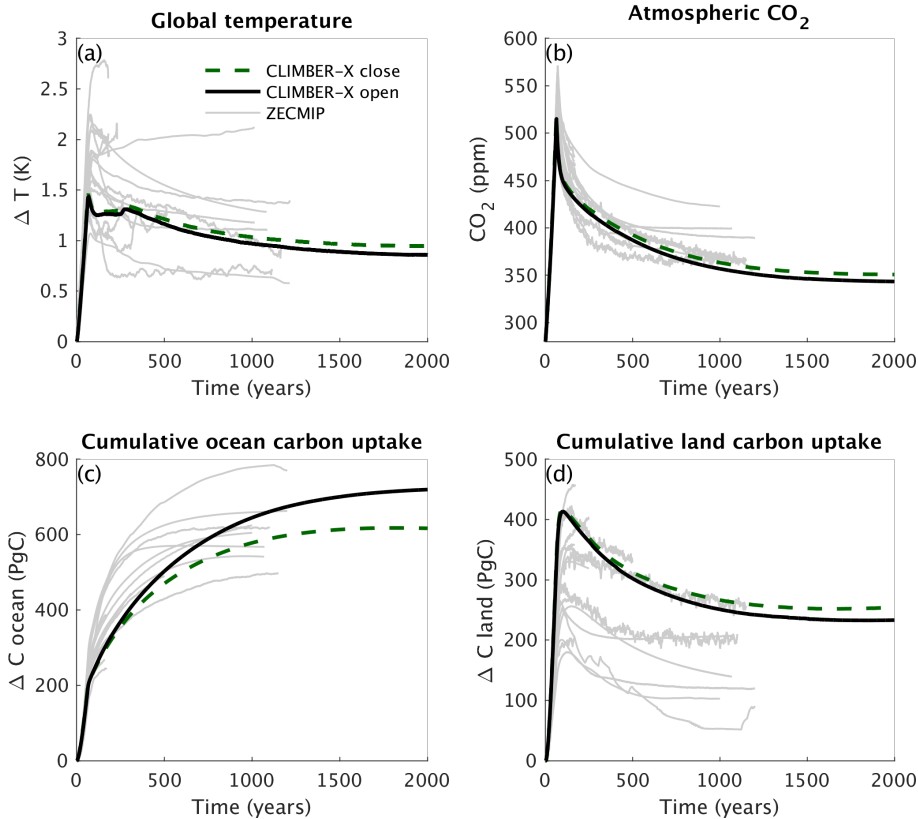

**Figure 34.** Comparison of CLIMBER-X simulations with ZECMIP model results in terms of (a) global temperature, (b) atmospheric $CO_2$ concentration, (c) cumulative ocean carbon uptake and (d) cumulative land carbon uptake for the standard ZECMIP experiment with 1000 PgC cumulative carbon emissions. For CLIMBER-X, results with both the open (solid lines) and closed (dashed lines) carbon cycle setups are shown.

to deficiencies in the simulated ocean circulation changes, which can at least partly be attributed to the comparatively coarse resolution of the ocean model and to the frictional-geostrophic approximation that it rests upon. On land, the carbon in soil layers below 1 m is underestimated, particularly in the permafrost zone, with possible implications for the land carbon cycle response to global warming.

Some possible directions for future model developments include the extension of the organic carbon cycle allowing for burial on land and in sediments and fluxes from land to ocean, the refinement of the carbonate chemistry on shelves, including coral growth, and possibly the addition of the nitrogen cycle on land, which could be important for nitrogen limitation of photosynthesis and would allow to have interactive atmospheric $N_2O$, considering that $N_2O$ fluxes from the ocean are already available from the ocean biogeochemistry model HAMOCC.

The computational efficiency of CLIMBER-X will enable it to be used for systematic explorations of the coupled climate-carbon cycle evolution over timescales ranging from decades up to ~100,000 years, while also allowing a quantification of related uncertainties.

*Code and data availability.* The source code of CLIMBER-X v1.0 is archived on Zenodo (https://doi.org/10.5281/zenodo.7898797), with
the exception of the HAMOCC model, which is covered by the Max Planck Institute for Meteorology software licence agreement as part of the MPI-ESM (https://code.mpimet.mpg.de/attachments/download/26986/MPI-ESM_SLA_v3.4.pdf). CMIP6 model data are licensed under a Creative Commons Attribution-ShareAlike 4.0 International License (https://creativecommons.org/licenses) and can be accessed through the ESGF nodes (for instance esgf-data.dkrz.de/search/cmip6-dkrz/).

*Author contributions.* M.W. and A.G. designed the model. T.I. and B.L. provided the HAMOCC model code and assisted in the imple-
mentation of the model into CLIMBER-X. C.H. helped with the sediment model setup and configuration. M.P. re-arranged HAMOCC into a column model for the purpose of integration into CLIMBER-X. M.H. implemented and tuned the particle ballasting scheme. D.D. contributed to the improvements in the land carbon cycle model. V.B. and G.M. assisted in the implementation and setup of the open carbon cycle. J.B., J.H. and G.R.M. developed the weathering model and contributed to its implementation into CLIMBER-X. M.W. coupled the different model components and tuned and tested the model. M.W. performed the model simulations, prepared the figures and wrote the
paper, with contributions from all authors.

*Competing interests.* The authors declare that they have no conflict of interest.

*Acknowledgements.* M.W., B.L., M.H. and J.B. are funded by the German climate modeling project PalMod supported by the German Federal Ministry of Education and Research (BMBF) as a Research for Sustainability initiative (FONA) (grant nos. 01LP1920B, 01LP1917D, 01LP1919B, 01LP1919C, 01LP1920C). G.M. is a Research Associate with the Belgian Fund for Scientific Research – F.R.S.-FNRS. We
thank Irene Stemmler for technical support in implementing HAMOCC into CLIMBER-X and Thomas Kleinen for discussions on the methane cycle. We acknowledge the World Climate Research Programme, which, through its Working Group on Coupled Modelling, coordinated and promoted CMIP6. We thank the climate modeling groups for producing and making available their model output, the Earth System Grid Federation (ESGF) for archiving the data and providing access, and the multiple funding agencies who support CMIP6 and ESGF. We thank Nuno Carvalhais for providing the soil and vegetation carbon dataset. The authors gratefully acknowledge the European
Regional Development Fund (ERDF), the German Federal Ministry of Education and Research and the Land Brandenburg for supporting this project by providing resources on the high performance computer system at the Potsdam Institute for Climate Impact Research.

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
