# Peer review of "The Earth system model CLIMBER-X v1.0 – Part 2: The global carbon cycle"

_Geoscientific Model Development, 2022_

## Referee Comment (RC1)

Review of The Earth system model CLIMBER-X v1.0 – Part 2: The global carbon cycle

By Matteo Willeit et al.

In this paper, Willeit et al detail the implementation of the carbon cycle in the new CLIMBER-X model. They explain the choices made in the atmosphere, ocean, land, and sediments and present results for the modern and historical periods, comparing with existing data. CLIMBER-X is a very promising model. Its rapidity is an asset, as well as the inclusion of many processes relevant for long-time scales, making it a very comprehensive model and a very good alternative to box models to study changes over periods longer than a few thousand years and up to a million years, a deed that more complex models are not capable of.

The paper is well organised and the model description and results well presented. However, the result section remains very descriptive, and it would greatly improve the paper to have more explanation and discussion on differences between model and data, and their potential causes. The simplifications and low resolution of CLIMBER-X are necessary to make it very efficient, but it is also important to have an idea of the limitations, their causes, and the possibilities for future improvements.

l. 29. There is a typo problem in the reference to "Raymo M.E. and Ruddiman W.F., 1992", it should read: "Raymo and Ruddiman, 1992"

Figure 1 (and Figure 3). For the molecules, could you put the numbers of atoms in subscript, such as for CO2, O3… -> $CO_2$, $O_3$…

l.111-112. Why have you modified the stochiometric ratio?

l. 127-128. Could you specify the variables involved for photosynthesis?

l.142. in "The rubisco-limited photosynthesis rate the version of PALADYN model": is there a missing "in"? -> The rubisco-limited photosynthesis rate **in** the version of PALADYN model

l. 278-280. what is the impact of the simplification?

Section4.1. When discussing the global values of Table 1 or 2 it would be useful to give the numbers in the text.

Table 1. The legend could be slightly expanded, e.g. Global values of the main ocean biogeochemical variables for the present-day.

l. 299. which models from CMIP6 are included?

l. 314-315. Why is the NPP in the Southern Ocean lower in climber-X ?

Figure 6. The zonal profiles are presumably global, are there differences between basins?

Figure 7. Since P is not limiting anywhere, could it be removed from the colorbar?

Figure 8. a,b,c,d are missing. Dust deposition is high everywhere in Africa in CLIMBER-X, especially in the southern part of Africa contrary to other models where the values are lower in the southern part compared to the northern part of Africa: why?

Figure 9. You could change the x and y labels, e.g. "modelled dust deposition" or "dust deposition in the model", and for the y axis: "observed dust deposition" or "dust deposition in observations".

l.334-l.335 what is the reference for the CMIP6 surface iron concentration? Could it be added to Figure 10?

l. 351-353 and Table 1. The surface silicate values are much higher than in the observation: why?

l.364-366. There are large biases in the $\delta^{13}C$ distribution: why?

l. 368-369. How are the winds in the model? Are they indeed underestimated? If the winds were higher and the mixing enhanced, would it not change also the nutrient distribution, which seems relatively good?

What is the explanation for the biases in the Southern Ocean for $\delta^{13}C$ and $\Delta^{14}C$?

l. 371 Why is there too little $CaCO_3$ in the Pacific?

l. 373 (and l.377) The table is referred to as "Tab.1" but before in the text (l.304) it has been referred to as "Table1".

l.374-375 Why is the opal content in sediments overestimated?

Table 2. As for table 1, the legend could be expanded, e.g. Global values for the main variables of the land carbon cycle.

l.385 Table 2 is referred to as "Tab.2", but "Table2" l.383.

l. 390-393. In Table 2, the data for soil carbon indicate a mean value of 4000 GtC , while CLIMBER-X simulates 2187 GtC, this seems like a large difference, could you discuss it?

l.395 Table 2 indicate that the carbon stock in permafrost has a mean value of 1300 Gtc, but CLIMBER-X simulated only 861 GtC: why is there so little carbon in permafrost? Is the permafrost area not large enough or the carbon storage too small? Are there missing processes? Or is it due to climate biases?

L419. Why is the silicate weathering low compared to data while the carbonate weathering is relatively high?

Section 4.2. Does it make a difference if you simulate the evolution during the historical period in the closed or open configurations? Presumably the impact of long-term processes on such a short period should be very limited but have you tried to simulate both to test this?

Conclusions. Could you discuss the biases, their potential causes, and what could be done to improve the model?

---

## Author Comment (AC1)

**Reviewer #1**

*In this paper, Willeit et al detail the implementation of the carbon cycle in the new CLIMBER-X model. They explain the choices made in the atmosphere, ocean, land, and sediments and present results for the modern and historical periods, comparing with existing data. CLIMBER-X is a very promising model. Its rapidity is an asset, as well as the inclusion of many processes relevant for long-time scales, making it a very comprehensive model and a very good alternative to box models to study changes over periods longer than a few thousand years and up to a million years, a deed that more complex models are not capable of.*
*The paper is well organised and the model description and results well presented. However, the result section remains very descriptive, and it would greatly improve the paper to have more explanation and discussion on differences between model and data, and their potential causes. The simplifications and low resolution of CLIMBER-X are necessary to make it very efficient, but it is also important to have an idea of the limitations, their causes, and the possibilities for future improvements.*

We would like to thank the reviewer for the constructive comments and suggestions on our paper. Below we provide a point-by-point response to the individual comments.

*l. 29. There is a typo problem in the reference to "Raymo M.E. and Ruddiman W.F., 1992", it should read: "Raymo and Ruddiman, 1992"*

Fixed.

*Figure 1 (and Figure 3). For the molecules, could you put the numbers of atoms in subscript, such as for CO2,O3... -> $CO_2$, $O_3$...*

Done.

*l.111-112. Why have you modified the stochiometric ratio?*

We have slightly increased the stochimetric Fe:C ratio to align with higher values used in other models. The effect of the larger Fe:C ratio is a stronger iron limitation of primary production, as more iron is needed to fix the same amount of carbon. Following the suggestion by reviewer #2 we have added further details on the iron cycle in the model to an Appendix.

*l. 127-128. Could you specify the variables involved for photosynthesis*

Photosysnthesis depends on absorbed shortwave radiation, air temperature, vapor pressure deficit between leaf and ambient air, atmospheric $CO_2$ and soil moisture. We have added this information to the paper.

*l.142. in "The rubisco-limited photosynthesis rate the version of PALADYN model": is there a missing "in"? -> The rubisco-limited photosynthesis rate in the version of PALADYN model*

Fixed.

*l. 278-280. what is the impact of the simplification?*

The impact of organic carbon fluxes with associated nutrients from land to the ocean and organic carbon burial in sediments represent an additional level of complication with relatively little observational constraints. We therefore decided to leave this for a separate study.

*Section4.1. When discussing the global values of Table 1 or 2 it would be useful to give the numbers in the text.*

Where appropriate, we added some numbers in the discussion when referring to the Tables 1 and 2.

*Table 1. The legend could be slightly expanded, e.g. Global values of the main ocean biogeochemical variables for the present-day.*

Done. Thanks for the suggestion.

*l. 299. which models from CMIP6 are included?*

The following CMIP6 models are included for ocean biogeochemistry: CESM2, IPSL-CM6A-LR, MRI-ESM2-0, MIROC-ES2L, MPI-ESM1-2-LR, UKESM1-0-LL and CanESM5.
For the land carbon cycle, the following models are used: ACCESS-ESM1-5, BCC-CSM2-MR, CanESM5, CNRM-ESM2-1, GFDL-ESM4, IPSL-CM6A-LR, MIROC-ES2L, MPI-ESM1-2-LR, MRI-ESM2-0, NorESM2-LM, UKESM1-0-LL.
We added this to the revised paper.

*l. 314-315. Why is the NPP in the Southern Ocean lower in climber-X ?*

NPP in the Southern Ocean is lower in CLIMBER-X than in MPI-ESM, but the MPI-ESM seems to be an outlier in this respect, possibly because of biases in climate, which have little to do with the HAMOCC ocean carbon cycle model. We will have a closer look at the physical climate biases in MPI-ESM and CLIMBER-X in the Southern Ocean.

*Figure 6. The zonal profiles are presumably global, are there differences between basins?*

We now make it clear in the caption that what is plotted are 'global zonal means'.
Obviously, we would expect some differences between different basins, but we think that global zonal means are sufficient for the purpose of a general comparison between different models.

*Figure 7. Since P is not limiting anywhere, could it be removed from the colorbar?*

We removed P from the colorbar.

*Figure 8. a,b,c,d are missing. Dust deposition is high everywhere in Africa in CLIMBER-X, especially in the southern part of Africa contrary to other models where the values are lower in the southern part compared to the northern part of Africa: why?*

We added a,b,c,d to the figure panels. Meanwhile, we improved performance of our dust module: the dust emission pattern changed substantially after the introduction of a topographic erodibility factor for dust emissions following Ginoux et al., 2001. The dust deposition fields are now generally in better agreement with other models and with observations with a notable improvement also over southern Africa.

*Figure 9. You could change the x and y labels, e.g. "modelled dust deposition" or "dust deposition in the model", and for the y axis: "observed dust deposition" or "dust deposition in observations".*

We changed the axes labels as suggested. We also fixed an error which resulted in the x- and y-axis labels being exchanged.

*l.334-l.335 what is the reference for the CMIP6 surface iron concentration? Could it be added to Figure 10?*

The comparison is referring to Fig. 11d, but a reference to the figure was missing in the text. We fixed that.

*l. 351-353 and Table 1. The surface silicate values are much higher than in the observation: why?*

A HAMOCC retuning, and in particular an increase in the Si/C uptake ratio in diatoms, substantially improved the simulated silicate values in general and especially at the surface.

*l.364-366. There are large biases in the δ13C distribution: why?*

The negative biases at 500-1500m depth are associated with the "nutrient trapping" problem (Aumont et al., 1999, Dietze and Loeptien, 2013) that is often seen in ESMs. This problem is characterised by high concentrations of remineralised nutrients and carbon and, therefore, low d13C (Liu et al. 2021). The positive biases are likely to result from too strong ventilation in the model. We added this discussion to the revised manuscript.

*l. 368-369. How are the winds in the model? Are they indeed underestimated? If the winds were higher and the mixing enhanced, would it not change also the nutrient distribution, which seems relatively good?*
*What is the explanation for the biases in the Southern Ocean for δ13C and Δ14C?*

The simulated average winds are not underestimated. This is more about non-linearity effects of synoptic variability. A one time mixing down to e.g. 100 m by a wind storm could/would have a large effect for some tracers. We would expect this non-linear effect to be much more important for radiocarbon than for nutrients.
See response to comment above for carbon isotope biases.

*l. 371 Why is there too little CaCO3 in the Pacific?*

The underestimation of calcite weight fractions in sediments of the eastern South Pacific Ocean is caused by water being undersaturated with respect to calcite in this area. This leads to dissolution of most of the calcite produced at the surface before it can even reach the sediments. The strongly undersaturated water is ultimately a result of deficiencies in the simulated ocean circulation. Some other models show similar deficiencies in the simulated calcite fraction in Pacific sediments (e.g. Kurahashi-Nakamura et al. 2020).

*l. 373 (and l.377) The table is referred to as "Tab.1" but before in the text (l.304) it has been referred to as "Table1".*

Fixed.

*l.374-375 Why is the opal content in sediments overestimated?*

After HAMOCC retuning, the global Opal fluxes to the sediment and the burial fluxes are perfectly within the range of recent estimates and the overestimation of opal content in the sediments is somewhat reduced. However, Opal is still too high in some regions. The model still overestimates Opal concentration in the eastern equatorial Pacific, simply as a result of missing CaCO3 in the sediments in that area. Opal sediment content is also overestimated in the northern Atlantic, for reasons that will be explored further. We also updated the observations of sediment content to the more recent dataset from Hayes et al. 2021, which provides an internally consistent analysis of CaCO3, Opal and TOC content in the surface sediment.

*Table 2. As for table 1, the legend could be expanded, e.g. Global values for the main variables of the land carbon cycle.*

Done. Thanks for the suggestion.

*l.385 Table 2 is referred to as "Tab.2", but "Table2" l.383.*

Fixed.

*l. 390-393. In Table 2, the data for soil carbon indicate a mean value of 4000 GtC , while CLIMBER-X simulates 2187 GtC, this seems like a large difference, could you discuss it?*

The simulated total soil carbon in the top soil meter in CLIMBER-X is well in the range of observational estimates, as shown in Table 2, indicating that the mismatch in total soil carbon content originates from un underestimation of carbon in deeper soil layers. One possible explanation for that is that the maximum turnover time scale of soil carbon is set to 5000 years in the model, which will limit the amount of carbon that can be accumulated. Other possible reasons include: (i) a general underestimation of vertical carbon transport by diffusion, particularly into perennially frozen soil layers, (ii) a possible depth dependence of soil carbon turnover due to processes other than temperature and moisture (e.g. Koven et al. 2013), which is not included in the model.
However, it should also be noted that carbon content in deeper soil layers is highly uncertain (e.g. Table 2 in Fan et al. 2020).
We added some of this discussion to the paper.

*l.395 Table 2 indicate that the carbon stock in permafrost has a mean value of 1300 Gtc, but CLIMBER-X simulated only 861 GtC: why is there so little carbon in permafrost? Is the permafrost area not large enough or the carbon storage too small? Are there missing processes? Or is it due to climate biases?*

This is due to an underestimation of carbon in deep soil layers, related to what discussed in the response to the point above. As shown in Table 2, the permafrost area is well captured by the model.

*L419. Why is the silicate weathering low compared to data while the carbonate weathering is relatively high?*

There was a bug in the model. The activation energy for silicates was used for carbonate weathering and viceversa. Since the activation energy for carbonates is lower than that for silicates, this resulted in an overestimation of carbonate and underestimation of silicate weathering fluxes. After the bugfix the weathering fluxes are well within the range of observations.

*Section 4.2. Does it make a difference if you simulate the evolution during the historical period in the closed or open configurations? Presumably the impact of long-term processes on such a short period should be very limited but have you tried to simulate both to test this?*

For the historical period it does not make any difference whether the closed or open carbon cycle are used. As seen e.g. from Fig. 32, the differences become tangible on millennial time scales.

*Conclusions. Could you discuss the biases, their potential causes, and what could be done to improve the model?*
Biases is the simulated distribution of ocean biogeochemical tracers exist and can mostly be related to deficiencies in the simulated ocean circulation changes, which can at least partly be attributed to

relatively coarse ocean model resolution and the frictional-geostrophic approximation employed in the 3D GOLDSTEIN ocean model. On land, the carbon in soil layers below 1 m is underestimated, particularly in the permafrost zone, with possible implications for the land carbon cycle response to global warming.

We added these points to the conclusion.

Additionally, following also a suggestion by Reviewer #2, we added a discussion on possible future directions for model development and improvements.

**Bibliography**

Aumont, O., Orr, J. C., Monfray, P., Madec, G., & Maier-Reimer, E. (1999). Nutrient trapping in the equatorial Pacific: The ocean circulation solution. *Global Biogeochemical Cycles*, *13*(2), 351–369. https://doi.org/10.1029/1998GB900012

Dietze, H., & Loeptien, U. (2013). Revisiting "nutrient trapping" in global coupled biogeochemical ocean circulation models. *Global Biogeochemical Cycles*, *27*(2), 265–284. https://doi.org/10.1002/gbc.20029

Ginoux, P., Chin, M., Tegen, I., Prospero, J. M., Holben, B., Dubovik, O., & Lin, S. J. (2001).Sources and distributions of dust aerosols simulated with the GOCART model. Journal of Geophysical Research Atmospheres, 106(D17), 20255–20273. https://doi.org/10.1029/2000JD000053

Koven, C. D., Riley, W. J., Subin, Z. M., Tang, J. Y., Torn, M. S., Collins, W. D., Bonan, G. B., Lawrence, D. M., & Swenson, S. C. (2013). The effect of vertically resolved soil biogeochemistry and alternate soil C and N models on C dynamics of CLM4. *Biogeosciences*, *10*(11), 7109–7131. https://doi.org/10.5194/bg-10-7109-2013

Hayes, C. T., Costa, K. M., Anderson, R. F., Calvo, E., Chase, Z., Demina, L. L., Dutay, J. C., German, C. R., Heimbürger-Boavida, L. E., Jaccard, S. L., Jacobel, A., Kohfeld, K. E., Kravchishina, M. D., Lippold, J., Mekik, F., Missiaen, L., Pavia, F. J., Paytan, A., Pedrosa-Pamies, R., … Zhang, J. (2021). Global Ocean Sediment Composition and Burial Flux in the Deep Sea. *Global Biogeochemical Cycles*, *35*(4), 1–25. https://doi.org/10.1029/2020GB006769

Kurahashi-Nakamura, T., Paul, A., Munhoven, G., Merkel, U., & Schulz, M. (2020). Coupling of a sediment diagenesis model (MEDUSA) and an Earth system model (CESM1.2): A contribution toward enhanced marine biogeochemical modelling and long-term climate simulations. *Geoscientific Model Development*, *13*(2), 825–840. https://doi.org/10.5194/gmd-13-825-2020

Liu, B., Six, K. D., & Ilyina, T. (2021). Incorporating the stable carbon isotope 13C in the ocean biogeochemical component of the Max Planck Institute Earth System Model. *Biogeosciences*, *18*(14), 4389–4429. https://doi.org/10.5194/bg-18-4389-2021

---

## Author Comment (AC2)

**Reviewer #2**

*The authors present the biogeochemical cycle implemented in CLIMBER-X. This model is one of the few models of intermediate complexity to simulate Earth system changes over many thousands of years. The model includes modules to simulate marine biogeochemical tracers, ocean-sediment interactions, and weathering as well as representation of the land biosphere, methane, and wetlands. This publication is timely and will serve the group and the community as a reference. The manuscript is generally clear and the authors manage to present a rich set of model outcomes. I recommend publications after the following comments have been addressed.*

We would like to thank the reviewer for the constructive comments and suggestions on our paper. Below we provide a point-by-point response to the individual comments.

*Representing ventilation time scale reasonably well is a prerequisite for simulating biogeochemical tracers in the ocean. Natural and bomb-produced radiocarbon, and CFCs are often used to probe the ventilation time scales of the ocean. The authors should present how well these different age tracers are simulated by the model.*

We followed the reviewer suggestion and expanded the discussion on ventilation ages, including also CFCs, as further outlined below.

*a) Results for CFC-11 and CFC-12 are not shown and discussed though mentioned in the model evaluation section 4 on line 290. Please compare simulated versus measured CFC-11 and CFC-12 distributions and inventories in additional figures and text.*

In the original paper version we did not included CFCs because they are not really part of the global carbon cycle. However, CFC-11 and CFC-12 tracers are included in the ocean model following the OMIP protocol and, as suggested by the reviewer, in the revised manuscript we added figures comparing CFCs distributions to observations.

*b) The marine radiocarbon distribution is presented at the very end of the ocean section (Fig. 20). However, age biases are key to understanding the biases in the various biogeochemical tracers. Please show and discuss radiocarbon results before the discussion of the other tracers.*

We followed the reviewer suggestion and now discuss age biases first.

*c) Please present and discuss the distribution of bomb-produced and natural radiocarbon separately*

We think that adding this additional analysis would not add substantial new information on the model performance, compared to existing analyses including CFCs.

*d) The discussion of biases in biogeochemical tracers (DIC, ALK, P, O2, Si, d3C…) would benefit by linking these biases to the biases in the age tracers.*

In the revised version of the paper we better link biases in the age tracers with the biases in other biogeochemical tracers.

*I miss a table that provides the model parameters. It would be very helpful if model parameters and key equations were summarized in a table (could also be in an appendix or as SI)*

We added an Appendix describing the model equations that were modified compared to the original description papers. We also summarized the relevant parameters in a Table.

*Iron limitation is prominent in the model (Fig.7). The authors should provide more detail on the iron cycle implemented in the model. How is the ratio of aeolian input versus advection of Fe in the Southern Ocean and elsewhere? How do the different iron sources of the model ocean compare with each other and other estimates? What is the role of ligands? Which parameters have been applied? What fraction of deposited iron is bioavailable? How will the balance between aeolian versus marine sources affect the model's sensitivity to glacial-interglacial dust deposition? Part of this information could be nicely incorporated in table 1.*

More details on the iron cycle are now given in the Appendix describing the key model equations.

*The model applies a temperature-sensitive particle remineralization rate. However, viscosity and thus particle sinking are also influenced by temperature. These two factors have opposite impacts under changing temperatures. Why is the temperature dependence of viscosity not considered? Will this cause a too-large sensitivity of atmospheric CO2 to changes in ocean temperature in glacial-interglacial simulations?*

A 10°C temperature increase roughly doubles the remineralization rate but changes water viscosity (and thus sinking velocity) by only 25%. We actually did test runs accounting for the viscosity effect on sinking velocity by multiplying it by a factor reference_viscosity/viscosity following equation 3 in Taucher et al. 2014 and we found that variable viscosity only caused an increase in atmospheric CO2 by about 2 ppm in LGM simulations, compared to expected >10 ppm CO2 decrease from the temperature-dependent remineralization rate (Ganopolski & Brovkin 2017). A further reason why we didn't include the viscosity effect is that the parameterization of the sinking speed is quite crude in the model. However, we are in the process of including the M4AGO particle sinking scheme (Maerz et al. 2020) in the model, which includes the viscosity effect on sinking speeds, and this may be included in a future release of CLIMBER-X.

*Conclusions: It would be nice if the authors present an outlook on future, planned (?) model improvements. For example, the implementations of flexible stoichiometry instead of constant Redfield ratios or N2O in the land biosphere appear to be possible targets.*

The main purpose of this paper is to present the current version of the CLIMBER-X model which is already used for various studies. In the future (as any other model) CLIMBER-X will be further developed. Some of possible future model developments could be:

- Organic carbon cycle with burial on land and in sediments, nutrient fluxes from land to ocean
- Include refined carbonate chemistry on shelfs, including corals
- Add nitrogen cycle on land: could be important for nitrogen limitation of photosynthesis and would allow to have interactive atmospheric N2O, because N2O fluxes from the ocean are already available from HAMOCC

*Title: The model remains an Earth System Model of Intermediate Complexity and this should be reflected in the title. Please replace the term Earth system model with Earth System Model of Intermediate Complexity.*

To be consistent with the title of the first part of CLIMBER-X description paper (Willeit et al. 2022) about the climate component, which is already published, we decided to leave the title as it is. We think that the first sentence in the abstract makes it sufficiently clear that it is an EMIC.

*L115: Please be more specific about how this virtual flux approach is implemented. Is the global net surface flux set to zero? How is the dilution effect implemented during times of net global freshwater addition/removal during ice sheet melt and formation?*

Two options are available in the model to implement the dilution effect on DIC and alkalinity in the case of the global net zero freshwater flux. The first one ensures that the net global surface tracer flux is zero by applying deviations from the global average freshwater flux to the global average surface tracer concentration. The second (default) option applies the actual local surface freshwater flux to compute a new virtual top ocean layer thickness and then dilutes the tracers accordingly. In this case, the conservation of tracer inventories is ensured by compensating disbalances over the global ocean.
Additionally, if the ocean volume is changing because of buildup or melt of land ice, concentrations of all tracers are globally adjusted while conserving tracer inventories. This is a reasonable simplification, considering that land ice volume changes occur on multi-millenial time scales, over which the ocean can be considered in approximate equilibrium.
This discussion has been added to the revised paper.

*L123: Could you please explain how this integration works? I guess HAMOCC in the ESM has a time step of order 30 min and is resolving daily radiation changes. How does the scheme account for sub-daily fluctuations in radiation for computing photosynthesis?*

The low-resolution version of the MPI-ESM model (MPI-ESM1.2-LR), for which HAMOCC was originally tuned, does not resolve the diurnal cycle of radiation in the ocean, but is tuned for daily mean radiation. This is therefore fully consistent with the HAMOCC implementation in CLIMBER-X. *"The MPI-ESM1.2-HR and MPI-ESM1.2-LR configurations differ in the atmosphere-ocean coupling frequency (section 2). Primarily, therefore, a different tuning of biological parameters in HAMOCC6 is required, as due to light limitation of plankton growth source-sink dynamics differ when using daily mean light versus considering a day-night cycle. The default tuning of the model assumes daily mean light."* /Mauritsen et al. 2019/

*L148: In CLM4.5 photosynthesis is downregulated by nitrogen limitation on a daily basis. This misconception/flawed approach has been corrected in CLM5. Why are you following this approach? It is strongly recommended to update to a more realistic N-limitation or is the model here running without N-limitation?*

The model is running without N-limitation, as there is no N cycle implemented. We simply use the formulation of CLM4.5 to compute the $V_{c,max}$, which involves a constant PFT-dependent value for the foliage nitrogen concentration.

*L174: Are there pools for products (paper, wood for construction) fed by deforestation? Please explain.*

The land use change scheme in the model is rather simple and does not include separate pools for products fed by deforestation. We have now added this clarification.

*L223: d13C of fossil fuel has changed considerably over the industrial period. Assuming a constant value of -26 permil leads to a positive bias. Please prescribe d13C of fossil emissions using updated information, e.g., Andres et al.*

As suggested by the reviewer, we have now introduced a variable d13C of fossil carbon emissions, prescribing it after Anders et al. 2016.

*L 293: Please comment on whether changes in C3 and C4 crops are prescribed for d13C discrimination.*

Following deforestation, the model will growth C3 or C4 grasses, depending on climate conditions.

*Fig. 2c I am a bit surprised that the DIC inventory at time 0, at the beginning of the spin-up is somewhat lower than the best estimate from the GLODAP data (37400 PgC), despite prescribing the DIC distribution from GLODAP at the beginning of the spin-up. Is the ocean volume too small?*

The discrepancy is a consequence of using a water density of 1000 kg/m3 when converting the GLODAP DIC concentrations in micro-mol/kg to kmol/m3, which is the unit for DIC in HAMOCC, during model initialisation. We have fixed this and now the actual 3D water density field is used instead.

*Fig. 13: Perhaps specify the boundary of the SO and whether the SO section is included in the profiles of the Atl., Pac., and Indian.*

The boundary of the Southern Ocean is set at 35°S and the Southern Ocean section is not included in the profiles of the Atlantic, Pacific and Indian ocean. We added this information to the figure caption.

*Figs. 14, 15, 17, 18, 19, 20: Why is the Southern Ocean sector only shown for the Indian Ocean? It would be much more instructive to show the SO sector of the Atlantic and Pacific in the top rows.*

We updated the figures as suggested.

*How are weathering fluxes distributed in the ocean?*

The weathering fluxes are transferred from the land to the ocean in the same way as water runoff, following the runoff routing scheme. We added this sentence to the paper.

**Bibliography**

Taucher, J., Bach, L. T., Riebesell, U., & Oschlies, A. (2014). The viscosity effect on marine particle flux: A climate relevant feedback mechanism. Global Biogeochemical Cycles, 28(4), 415–422. https://doi.org/10.1002/2013GB004728

Ganopolski, A., & Brovkin, V. (2017). Simulation of climate, ice sheets and CO2 evolution during the last four glacial cycles with an Earth system model of intermediate complexity. *Climate of the Past*, *13*(12), 1695–1716. https://doi.org/10.5194/cp-13-1695-2017

Maerz, J., Six, K. D., Stemmler, I., Ahmerkamp, S., & Ilyina, T. (2020). Microstructure and composition of marine aggregates as co-determinants for vertical particulate organic carbon transfer in the global ocean. Biogeosciences, 17(7), 1765–1803. https://doi.org/10.5194/bg-17-1765-2020

Mauritsen, T., Bader, J., Becker, T., Behrens, J., Bittner, M., Brokopf, R., Brovkin, V., Claussen, M., Crueger, T., Esch, M., Fast, I., Fiedler, S., Fläschner, D., Gayler, V., Giorgetta, M., Goll, D. S., Haak, H., Hagemann, S., Hedemann, C., … Roeckner, E. (2019). Developments in the MPI-M Earth System Model version 1.2 (MPI-ESM1.2) and Its Response to Increasing CO2. *Journal of Advances in Modeling Earth Systems*, *11*(4), 998–1038. https://doi.org/10.1029/2018MS001400